# Light-mediated discovery of surfaceome nanoscale organization and intercellular receptor interaction networks

Maik Müller [1,2], Fabienne Gräbnitz[3], Niculò Barandun[3], Yang Shen [4], Fabian Wendt [1,2], Sebastian N. Steiner [1,2], Yannik Severin [5], Stefan U. Vetterli [6], Milon Mondal [6], James R. Prudent [7], Raphael Hofmann [8], Marc van Oostrum [1,2], Roman C. Sarott [8], Alexey I. Nesvizhskii [9,10], Erick M. Carreira [8], Jeffrey W. Bode [8], Berend Snijder [5,2], John A. Robinson [6], Martin J. Loessner [4], Annette Oxenius [3] & Bernd Wollscheid [1,2✉]

The molecular nanoscale organization of the surfaceome is a fundamental regulator of cellular signaling in health and disease. Technologies for mapping the spatial relationships of cell surface receptors and their extracellular signaling synapses would unlock theranostic opportunities to target protein communities and the possibility to engineer extracellular signaling. Here, we develop an optoproteomic technology termed LUX-MS that enables the targeted elucidation of acute protein interactions on and in between living cells using light-controlled singlet oxygen generators (SOG). By using SOG-coupled antibodies, small molecule drugs, biologics and intact viral particles, we demonstrate the ability of LUX-MS to decode ligand receptor interactions across organisms and to discover surfaceome receptor nanoscale organization with direct implications for drug action. Furthermore, by coupling SOG to antigens we achieved light-controlled molecular mapping of intercellular signaling within functional immune synapses between antigen-presenting cells and CD8[+] T cells providing insights into T cell activation with spatiotemporal specificity. LUX-MS based decoding of surfaceome signaling architectures thereby provides a molecular framework for the rational development of theranostic strategies.

[1] Department of Health Sciences and Technology (D-HEST), ETH Zurich, Institute of Translational Medicine (ITM), Zurich, Switzerland. [2] Swiss Institute of Bioinformatics (SIB), Lausanne, Switzerland. [3] Department of Biology, ETH Zurich, Institute of Microbiology, Zurich, Switzerland. [4] Institute of Food Nutrition and Health, Department of Health Sciences and Technology, ETH Zurich, Zurich, Switzerland. [5] Institute of Molecular Systems Biology, Department of Biology, ETH Zurich, Zurich, Switzerland. [6] Chemistry Department, University of Zurich, Zurich, Switzerland. [7] Centrose LLC, Madison, WI, USA. [8] Laboratory of Organic Chemistry, Department of Chemistry and Applied Biosciences, ETH Zurich, Zurich, Switzerland. [9] Department of Pathology, University of Michigan, Ann Arbor, MI, USA. [10] Department of Computational Medicine and Bioinformatics, University of Michigan, Ann Arbor, MI, USA. ✉email: wbernd@ethz.ch

Cellular function is regulated by information exchange with the outside world. The cell surface proteotype (surfaceome) thereby acts as the signaling gateway to the cell by connecting external molecular cues with intracellular response pathways within dynamic surfaceome signaling domains[1,2]. The spatiotemporal organization of these structures ranging in size from ~100 nm (B-cell receptor (BCR) clusters) to several microns (immune synapses) essentially determines cellular phenotypes by regulating receptor signaling[3,4], drug response[5], host-pathogen interactions[6], and intercellular communication[7]. The tremendous therapeutic potential encoded in the surfaceome landscape catalyzes the emergence of innovative approaches that exploit receptor proximities for selective degradation of disease-causing receptors[8,9] and for the targeting of cells with unmatched precision using multi-specific small molecules[10], antibodies[11], extracellular-drug conjugates (EDC)[12] and colocalization-dependent protein switches[13]. The ability to decipher cell-type-specific surfaceome nanoscale organizations and ligand-targeted receptor microenvironments is, therefore, a prerequisite for the understanding of fundamental cellular signaling processes and for the rational design of next-generation precision medicines.

Still today, the surfaceome landscape remains terra incognita that cannot be inferred from bulk proteomic and transcriptomic measurements. Dedicated strategies have therefore been established to profile cellular surfaceomes. For example, the extensive application of the Cell Surface Capture (CSC) technology[14,15] established N-glycosylated surfaceomes for numerous cell types (collectively reported in the Cell Surface Protein Atlas, CSPA https://wlab.ethz.ch/cspa/)[16] and enabled the in silico characterization of the entire human surfaceome[17]. While, fluorescence-based reporter assays[18–20] and platforms of genetically engineered receptors[21–23] established associations between receptors by high-throughput testing of binary interactions within and across surfaceomes, the Ligand-Receptor Capture (LRC) technology enabled the identification of direct ligand interactions of N-glycosylated receptors[24,25]. Most recently, localized proximity labeling by antibody-tethered enzymes (APEX[26], horseradish peroxidase (HRP), PUP-iT[27]) or photocatalysts (μMap)[28] were successfully combined with mass spectrometry for the characterization of selected surfaceome landscapes including lipid rafts[29], BCR neighborhoods[30] and intact neuronal synapses[31] in living cells and in fixed tissues[32]. Although such antibody-based methods are useful in deciphering surface microenvironments of targeted receptors, highly versatile technologies are required to also uncover dynamic surfaceome domains that underlie the signaling function of small molecules, biomolecules, viral particles, and complex intercellular receptor interaction networks as formed within immunosynapses of interacting immune cells during T-cell activation.

Singlet oxygen generators (SOG) are well known for photo-catalytic generation of short-lived singlet oxygen (SO) that promiscuously oxidize biomolecules including proteins in nanometer vicinity[33,34]. The lifetime and traveling distance of photosensitized SO species thereby depend on environmental conditions and are drastically enhanced in heavy water ($D_2O$) based solutions[35–37]. Small-molecule SOG[38] and genetically encodable versions thereof are extensively used throughout life sciences for chromophore-assisted light inactivation of proteins[39–42] and cells[43,44], correlative light-electron microscopy[45], and the detection of intracellular as well as extracellular protein interactions[46–48]. Owing to the light-controlled activity, the tunable oxidation range, and the molecular size that is potentially compatible with any type of ligand, we recognized small molecular SOG as ideal probes for the in situ labeling of ligand-targeted surfaceome signaling domains of virtually any description.

Here, we developed a SOG-based and light-controlled proximity labeling technology termed LUX-MS enabling the cell type-specific elucidation of surfaceome nanoscale organization and ligand-targeted signaling domains on the surface of living cells of essentially any organism. The approach capitalizes on ligand-coupled small-molecule SOG for the specific and light-tunable biotinylation of ligand-proximal proteins enabling their identification using a high-throughput robot-assisted proteotyping workflow. We demonstrate the spatial specificity and broad applicability of our approach to reveal receptor microenvironments and surfaceome signaling domains of small molecules, biomolecules, and intact viruses in eukaryotic and prokaryotic systems in a discovery-driven fashion and without the need for genetic manipulation. At last, we highlight the utility of LUX-MS to elucidate intercellular surfaceome signaling domains of the highest complexity by mapping the architecture of functional immunosynapses in molecular detail.

## Results

**Development of the LUX-MS technology.** We first tested the capability of the small-molecule SOG thiorhodamine to photo-oxidize transferrin proteins in vitro. High-resolution liquid chromatography-tandem mass spectrometry (LC-MS/MS) analysis combined with open modification peptide searching revealed the light-dependent conversion of the amino acids histidine (His), cysteine (Cys), tryptophan (Trp), tyrosine (Tyr), and methionine (Met) into previously described photo-oxidation products[33] at varying $D_2O$ concentrations (Supplementary Fig. 1). Next to oxidized Trp and Tyr, the two most prevalently formed modifications His+14 and His+32, previously characterized as 2-oxo-histidine (2-imidazolone) and its hydrated form[33,49], respectively, bear light-induced ketone groups that potentially offer functionalization via hydrazone formation using hydrazide-containing linkers. Indeed, photo-oxidation of transferrin in the presence of a biotin-hydrazide (BH) linker led to significant consumption of both His+14 and His+32 (Supplementary Fig. 1) indicating biotinylation of photo-oxidized proteins via light-activated histidines. We then tailored a liquid-handling robot-assisted processing workflow to automatically capture and proteolytically digest biotinylated proteins within streptavidin-functionalized tips for LC-MS/MS-based analysis eventually enabling the discovery-driven identification and label-free quantification of labeled proteins with excellent reproducibility and at high sample throughput (96-well format)[14]. Based on this optoproteomic workflow, we designed the LUX-MS technology (Fig. 1) where SOGs are chemically coupled to any ligand of interest and guided to selected surfaceome signaling domains on living cells to enable their light-induced labeling under physiological conditions. Proteins of the in situ labeled microenvironment are then identified in a hypothesis-free fashion by light-dependent enrichment against a non-labeled control.

**LUX-MS enables proteome-wide mapping of antibody binding targets and the surfaceome nanoscale organization.** Surfaceome interactions determine antibody specificity and signaling capacity and their elucidation is critical for the development of therapeutic antibodies such as Rituximab, a highly successful therapeutic antibody for the treatment of B-cell malignancies[50]. Decades of research were required to eventually reveal the surface microenvironment of its primary receptor CD20 and its role in BCR-dependent cell killing[51]. To demonstrate the utility of LUX-MS to provide insights into therapeutic antibody action, we coupled anti-CD20 antibodies to the small-molecule SOG thiorhodamine with an average degree-of-labeling (DOL) of 1.5 SOG molecules per antibody (Fig. 2a and Supplementary Fig. 2). Incubating Ab-SOG constructs with freshly isolated human peripheral blood mononuclear cells (PBMCs) resulted in antibody binding to 8% of all detected cells (Fig. 2b) corresponding to the typical fraction

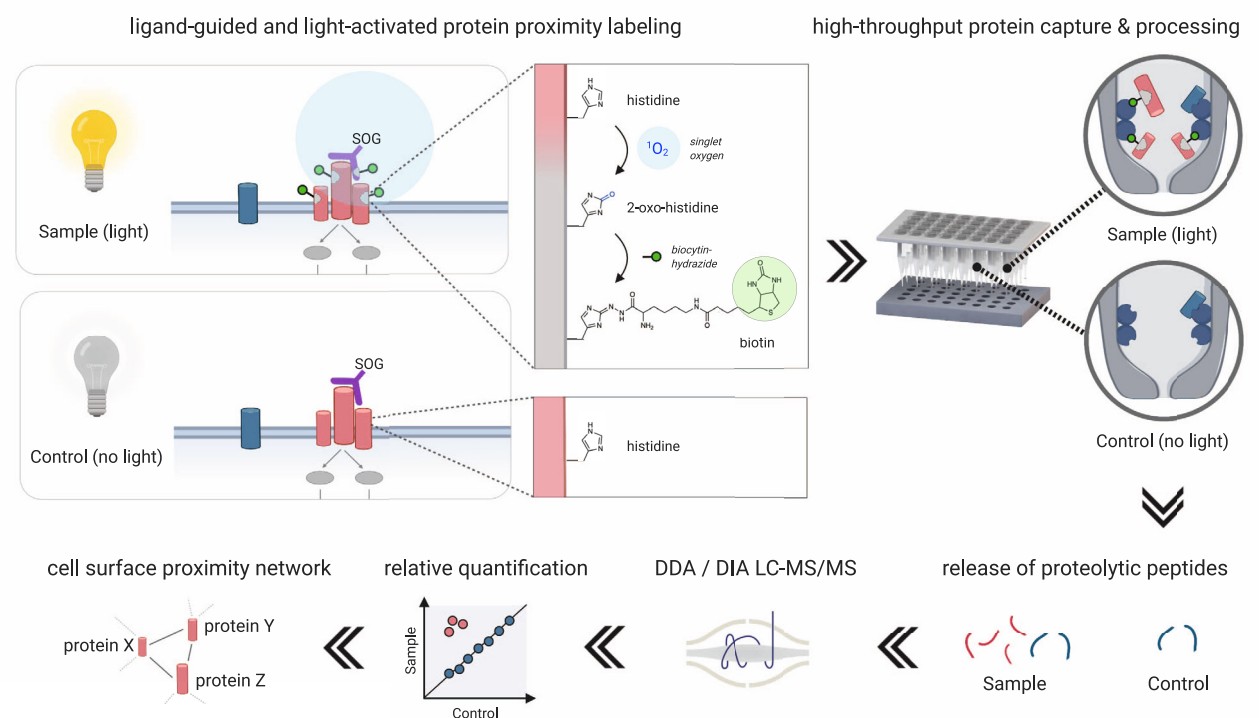

**Fig. 1 The LUX-MS strategy for the light-controlled discovery of surfaceome nanoscale organization and ligand-targeted signaling domains.** Small-molecule singlet oxygen generators (SOG) are chemically coupled to antibodies or ligands of any description (purple) and activated by visible light to establish spatiotemporally controlled oxidation of targeted surfaceome landscapes on living cells (red). Light-induced conversions of amino acids such as histidine into 2-oxo-histidine (gray) enable the stable biotinylation of SOG-proximal proteins under physiological conditions using biotin-functionalized hydrazide linkers (biocytin-hydrazide, green). In situ labeled proteins are automatically captured in streptavidin-functionalized tips and processed using a liquid-handling robotic platform ensuring fast and reproducible mass spectrometry-based quantification across a large number of samples. Relative protein quantification between labeled (light) and unlabeled (i.e., no light) samples ultimately enables discovery-driven and proteome-wide identification of cell surface proximity networks that constitute acute antibody- and ligand-targeted surfaceome signaling domains on living cells.

of CD20-positive cells in human PBMCs (5–10%)[52]. By applying light for 5 min in the presence of BH, 96% of all Ab-SOG bound but not unbound cells were surface biotinylated, demonstrating a proximity labeling reaction that is controllable by light and targetable to individual cells in a mixed cell population.

We then examined LUX-labeling dynamics using Ab-SOG on a human B-lymphoma cell line and observed light-dependent cell surface biotinylation to be fine-tunable by the duration of illumination (Fig. 2c and Supplementary Fig. 2). The extend of labeling was further tunable by modulating buffer conditions, i.e., increasing $D_2O$ concentration and pH in line with enhanced photo-oxidation (Supplementary Fig. 4) and histidine conversion[53], respectively. We then performed a CD20-targeted LUX-MS experiment by treating cultured human B-lymphoma cells with Ab-SOG prior illumination for 0–5 min. In total, 1674 proteins were quantified by LUX-MS with at least two unique proteotypic peptides per protein (Fig. 2d and Supplementary Fig. 2). Most proteins were equally abundant in the non-illuminated conditions, however, 63 proteins showed striking enrichment that correlated with the duration of illumination and that culminated in a fivefold abundance increase after 5 min of light. Of these, 84% were bona fide surface proteins while only 1% of all non-enriched proteins were surface associated (Fig. 2e). We thereby specifically retrieved 43 of the 215 CSPA-reported surfaceome members for this cell line including non-glycosylated surface proteins that typically evade CSC- or LRC-based surfaceome interrogations (Supplementary Fig. 3). LUX-MS, therefore, enables unbiased spatial proteotyping with sub-surfaceome resolution. In a second experiment using

replicates, we used statistical testing to identify bona fide SOG-proximal candidates (enrichment fold change >1.5 and $p$ value < 0.05, Fig. 2e and Supplementary Data 1). Among the top hits, we found immunoglobulin G2 (IgG2) isoforms of the Ab-SOG, the primary binding target CD20, human leukocyte antigen class II, tetraspanin CD37, CD298, and CD71 that are physically or functionally associated with CD20 on the surface of B-lymphoma cells[12,54–61]. Furthermore, LUX-MS revealed the proximity of CD20 to the IgM BCR complex (CD79A, CD79B, IgM heavy, and light chain) and BCR-associated CD19, CD38, and ITGA4 in line with a model where rituximab binding to CD20 is able to eradicate continuously activated B-lymphoma cells by altering survival-promoting BCR signaling[51]. On resting human B cells that express lower levels of CD20, we found CD20 in proximity to the BCR in presence of CD40 and partial absence of IgM indicating its segregation to quiescent IgD BCR nanoclusters as demonstrated previously[51] (Supplementary Fig. 4). Performing LUX-MS in $D_2O$ recapitulated these results and, in addition, provided CD20-proximity candidates that were not identified in $H_2O$ alone, thereby increasing the number of CD20-proximity candidates from 118 to 172 (Fig. 2d and Supplementary Fig. 4). In contrast, CD20-proximity labeling using HRP-coupled antibodies yielded 220 candidates with a partially overlapping set, including the target receptor and interactors. Furthermore, using SOG-coupled antibodies against CD38, CD54, CD166, and CD220 revealed distinct receptor-specific surfaceome neighborhoods that covered a wide range of protein functionalities (Fig. 2f). LUX-MS, therefore, enables the discovery of cell-type-specific surfaceome microenvironments of selected receptors on living cells with fine-tunable

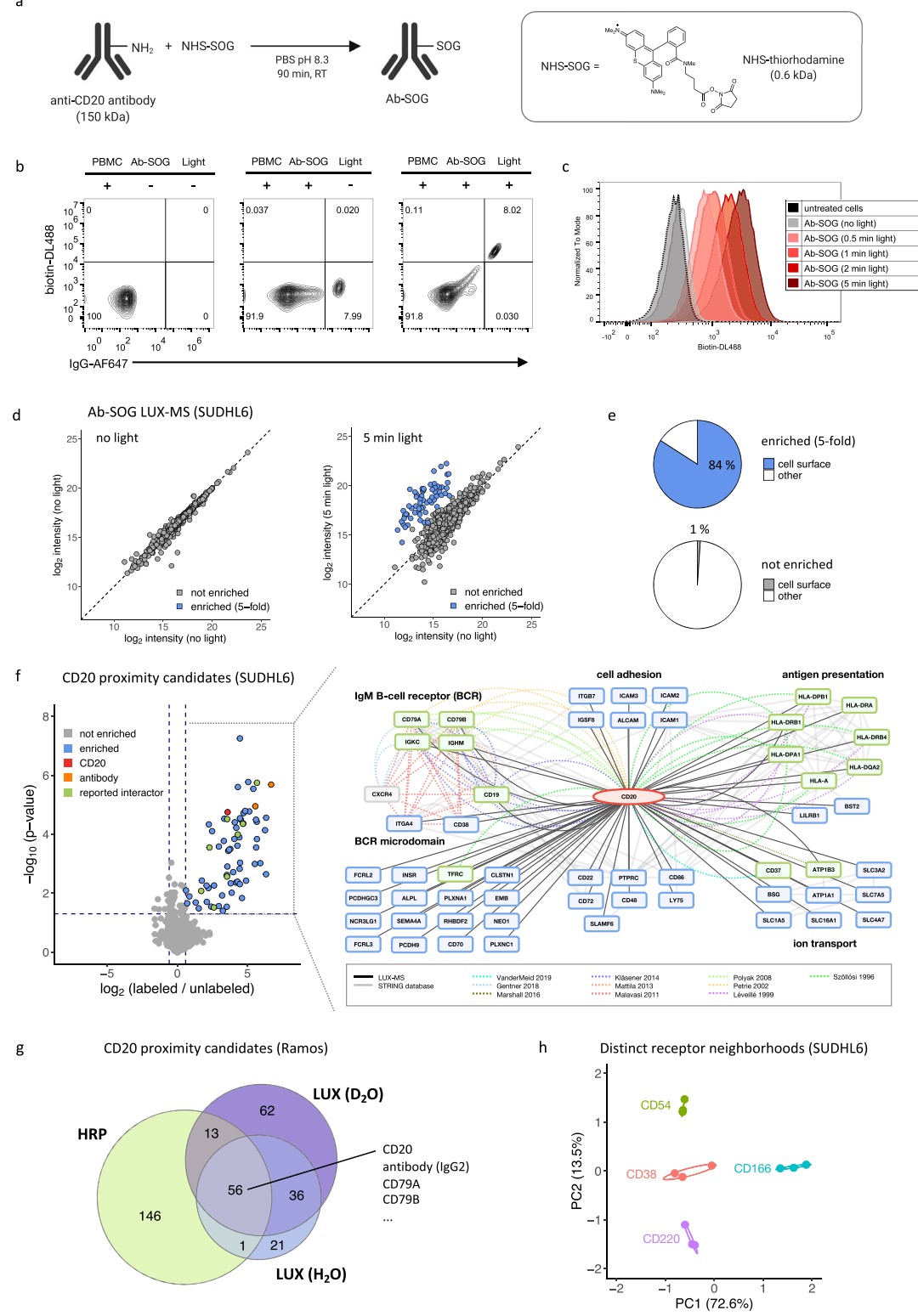

spatial precision and as such provides a means for the systematic mapping of the global surfaceome nanoscale organization of a cell on a proteome-wide scale.

**LUX-MS decodes surfaceome signaling domains of small-molecule drugs and biomolecules**. Next to therapeutic antibodies,

identifying receptors and surfaceome microenvironments targeted by small molecules is crucial for the development of small-molecule drugs and next-generation precision therapies such as drug-conjugated bispecific antibody (DBA)[11] and EDC technologies[12]. EDC exploit cell surface proximities between disease-relevant antibody targets and small-molecule payload receptors for efficient cell ablation. However, the identity of small-molecule receptors and their

**Fig. 2 Proteome-wide discovery of antibody binding targets and surfaceome nanoscale organization. a** Schematic of decorating an antibody (Ab) with the singlet oxygen generator (SOG) thiorhodamine. **b** Flow cytometric analysis of human peripheral blood mononuclear cells (PBMC) stained for cell surface biotin and immunoglobulin before and after LUX-labeling with anti-CD20 antibody-SOG conjugate (anti-CD20-SOG) (>60,000 cells per condition). **c** Histogram plot showing light-dependent cell surface biotinylation of LUX-labeled B-lymphoma SUDHL6 cells (>50,000 cells per condition). **d** Scatter plot showing light-dependent enrichment of LUX-MS quantified proteins from anti-CD20-SOG treated B-lymphoma SUDHL6 cells. **e** Fraction of surface proteins of LUX-MS quantified proteins found to be 5-fold enriched or not enriched after illumination of anti-CD20-SOG-treated B-lymphoma SUDHL6 cells. **f** Left, Volcano plot showing relative abundance changes of LUX-MS quantified proteins from anti-CD20-SOG treated B-lymphoma SUDHL6 cells with and without illumination for 5 min, tested using a two-sided Student's *t* test. Green and blue dots represent known and previously unknown CD20 associated proteins, respectively. Orange and red dots represent chains of the used antibody and the primary binding target CD20. Right, literature and LUX-MS-based cell surface interaction network of CD20. **g** Venn diagram showing overlap of CD20 proximal candidates identified on resting human B cells (Ramos) using horseradish peroxidase (HRP) conjugated antibodies or LUX-MS performed once in water ($H_2O$) or heavy water ($D_2O$) based buffer. **h** PCA analysis of CD38, CD54, CD166, and CD220 receptor microenvironments identified by antibody-guided LUX-MS on living B-lymphoma SUDHL6 cells. Source data are provided as a Source Data file and interactive volcano plots (Supplementary Data 1).

organization within the surfaceome of live cells often remains unknown as enzyme-based proximity detection strategies are not attuned to using small-molecule ligands. To test whether LUX-MS can reveal small-molecule-targeted surfaceome domains, the SOG thiorhodamine was covalently conjugated to CG1, a cytotoxic drug that selectively inhibits the active ion pumping subunit (ATP1A1) of the ubiquitously expressed human $Na^+/K^+$-ATPase complex (NKA) (Fig. 3a)[12]. A single-cell chemosensitivity screen using cultured human promyelocytic leukemia (HL60) cells showed potent cell ablation by both CG1 ($IC_{50} = 3.5$ nM) and CG1-SOG ($IC_{50} = 93$ nM) after 48 h, whereas the coupling of CG1 to HRP with an average DOL of 2.5 CG1 molecules per enzyme essentially abolished its drug activity (Fig. 3b and Supplementary Fig. 6). Flow cytometric analysis of HL60 cells LUX-labeled with increasing CG1-SOG concentrations showed light-induced and concentration-dependent cell-surface biotinylation (Supplementary Fig. 6). In a subsequent LUX-MS experiment, 48 CG1 proximity candidates were identified on the surface of HL60 cells (Fig. 3c and Supplementary Data 2) including the beta3 subunit of the NKA (ATP1B3), indicating that LUX-MS enables the discovery-driven identification of small-molecule receptors. Furthermore, we found the cell-surface receptor Basigin (CD147), which is a functional target of CG1-loaded EDCs on the cell type used for this experiment[12]. Therefore, the data suggest that small-molecule-targeted receptors and associated surfaceome signaling domains can be revealed by LUX-MS eventually enabling systematic target discovery for DBA or EDC development.

We next asked whether LUX-MS can be applied to uncover surfaceome interactions of more complex ligands, i.e., secreted biomolecules that are key elements of intercellular communication. We, therefore, coupled thiorhodamine SOG to the peptide hormone insulin and the plasma glycoprotein transferrin that regulate cellular homeostasis through interactions with insulin receptor or insulin-like growth factor 1 receptor (IGF1R) homo- or heterodimers, and the transferrin receptor (TFRC), respectively (Fig. 3d). LUX-MS was performed on human B-lymphoma cells that were treated with either conjugate and illuminated for 5 min. Results demonstrated specific enrichment of insulin or transferrin in conjunction with their cognate receptors INSR and IGF1R or TFRC, respectively (Fig. 3e and Supplementary Data 3). Transferrin-guided LUX-MS further revealed CD22, a B-cell specific lectin shown to weakly bind to sialylated *N*-glycoproteins including transferrin[62], and the cation-independent mannose-6-phosphate receptor M6PR, a sorting enzyme involved in the clathrin-mediated endocytosis and recycling of transferrin-bound receptors that was shown to co-localize with TFRC prior to internalization into sorting endosomes[63]. Thus, in addition to small molecules, LUX-MS can also be used to unravel surfaceome signaling domains of biomolecules providing molecular insights into long-range intercellular communication.

**Elucidation of surfaceome signaling interactions across domains of life**. Protein proximity labeling and detection by LUX-MS primarily rely on light-induced modifications of histidine, whose cellular biosynthesis is evolutionary well conserved[64]. Thus, LUX-MS may uncover ligand-targeted surfaceome signaling domains in non-eukaryotic systems such as bacteria where LRC-like approaches are not applicable, aiding in the further development of anti-infective agents such as Thanatin to fight the growing threat of drug-resistant bacteria. Thanatin is a naturally occurring peptide that was recently shown to inhibit the outer membrane biogenesis complex Lpt in *Escherichia coli* (*E. coli*) by binding to LptA[65]. To demonstrate the utility of LUX-MS to identify such important surfaceome interactions in bacteria, we coupled Thanatin to the SOG methylene blue (Fig. 4a) and added it to *E. coli* cells with or without competition by an excess of unconjugated Thanatin. All cells were subsequently illuminated for 5 min with red light at 656 nm and subjected to LUX-MS for identification of proximity labeled proteins. Numerous outer membrane and periplasmic proteins were significantly enriched (Fig. 4b and Supplementary Data 4), indicating membrane-localized labeling by SOG-coupled Thanatin. LUX-MS thereby revealed the functional binding partners of Thanatin LptA and LptD as top enriched hits and suggested proximity to outer membrane proteins including BamA and BamC (Fig. 4b). Indeed, the Bam complex was shown to catalyze the folding and insertion of outer membrane proteins including LptD forming supramolecular surfaceome clusters on living bacteria[66]. LUX-MS therefore enables deconvolution of ligand-targeted receptors and associated surfaceome microenvironments across domains of life.

**Unraveling the surfaceome interaction network of intact viral particles using LUX-MS**. In addition to drug-resistant bacteria, pathogenic viruses pose a growing threat to global human health. Extracellular host-pathogen interactions are thereby key for cellular tropism and infection and, as such, are primary research targets for the classification and diagnosis of infectious diseases and vaccine discovery. To demonstrate the utility of LUX-MS to reveal viral surfaceome interactions, we coupled thiorhodamine SOG to bacteriophages that recognize the ubiquitous cell wall component teichoic acid of Gram-positive *Listeria monocytogenes* (*Lm*), a highly infectious food-borne bacteria that causes listeriosis[67] (Fig. 5a). *Lm* was incubated with SOG-coupled phages at a multiplicity of infection (MOI) of 1 or 10 and LUX-labeled by illumination for 0–15 min. Flow cytometry analysis revealed the light-dependent cell-surface biotinylation of individual bacteria that reached saturation after 15 min of illumination at an MOI of 10 (Fig. 5b). Subsequent LUX-MS analysis revealed light-induced, significant enrichment of 146 proteins including 19 phage and 127 *Lm* proteins (Fig. 5c and Supplementary Data 5). We found components of bacteriophage capsid (Cps, Gp5, Gp9),

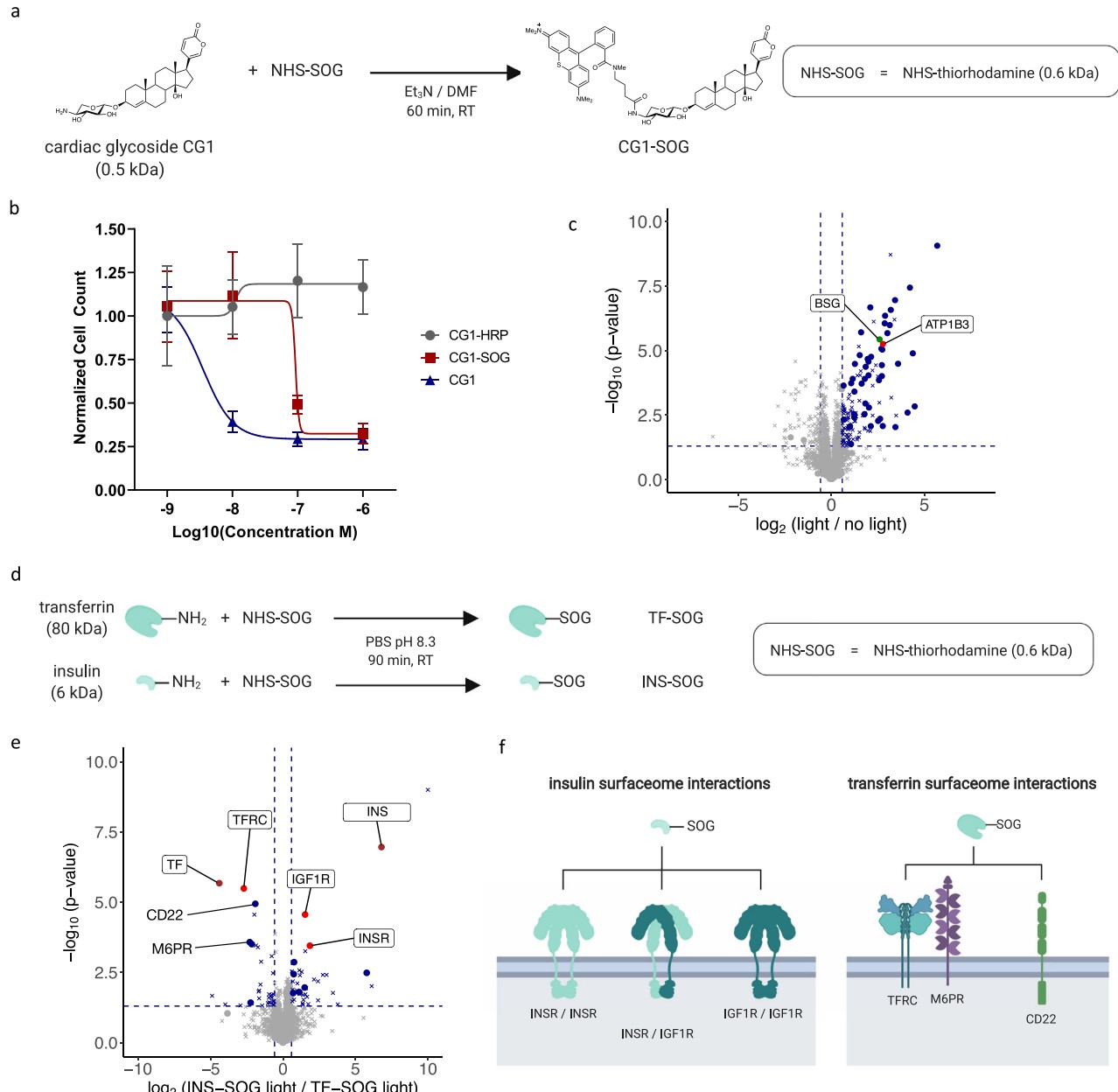

**Fig. 3 Decoding surfaceome signaling domains of small-molecule drugs and biomolecules. a** Schematic of coupling the small-molecule drug cardiac glycoside CG1 to the singlet oxygen generator (SOG) thiorhodamine. **b** Single-cell chemosensitivity screen showing the viability of human promyelocytic leukemia (HL60) cells after incubation with free CG1, CG1 coupled to thiorhodamine (CG1-SOG) or horseradish peroxidase (CG1-HRP) for 48 h. Data are presented as mean values ±SD ($n = 10$ technical replicates). **c** Volcano plot showing relative abundance changes of LUX-MS quantified proteins from CG1-SOG treated promyelocytic leukemia HL60 cells with and without illumination for 5 min, tested using a two-sided Student's $t$ test. Dots and crosses represent cell surface and otherwise annotated proteins, respectively. Red, green and blue dots represent the binding target of CG1, known and previously unknown surfaceome interactors, respectively. The former two are highlighted. **d** Schematic of coupling the biomolecules insulin and transferrin to the singlet oxygen generator (SOG) thiorhodamine. **e** Volcano plot showing relative abundance changes of LUX-MS quantified proteins from insulin-SOG and transferrin-SOG treated B-lymphoma SUDHL6 cells illuminated for 5 min, tested using a two-sided Student's $t$ test. Dots and crosses represent cell surface and otherwise annotated proteins, respectively. Brown, red and blue dots represent the ligand, primary binding target, and potential surfaceome interactors, respectively. The former two are highlighted together with known surfaceome interactors. **f** Schematic representation of the surfaceome proximity network identified by LUX-MS. Source data are provided as a Source Data file and interactive volcano plots (Supplementary Data 2 and 3).

sheath (Gp3), baseplate (Gp20), and tail (Gp17, Gp18, Gp19, Gp23, Tmp, Tsh) and enzymes involved in DNA binding (Gp37), replication (Gp45), recombination (Gp44), transcription (Gp7), and immunoprotective functions (Gp32) indicating thorough LUX-labeling of host cell-attached bacteriophages. For *Lm* proteins, gene ontology analysis revealed 81 extracellular and only 8 intracellular proteins (Fig. 5d) including major virulence factors such as cell wall-associated internalins A and B that mediate host cell invasion, membrane-localized ActA that governs cell-to-cell spreading, and ATP-binding cassette (ABC) transporters that mediate multi-drug resistance[68] (Fig. 5e). Using HA-tagging and immunofluorescence microscopy, we further validated the *Lm*

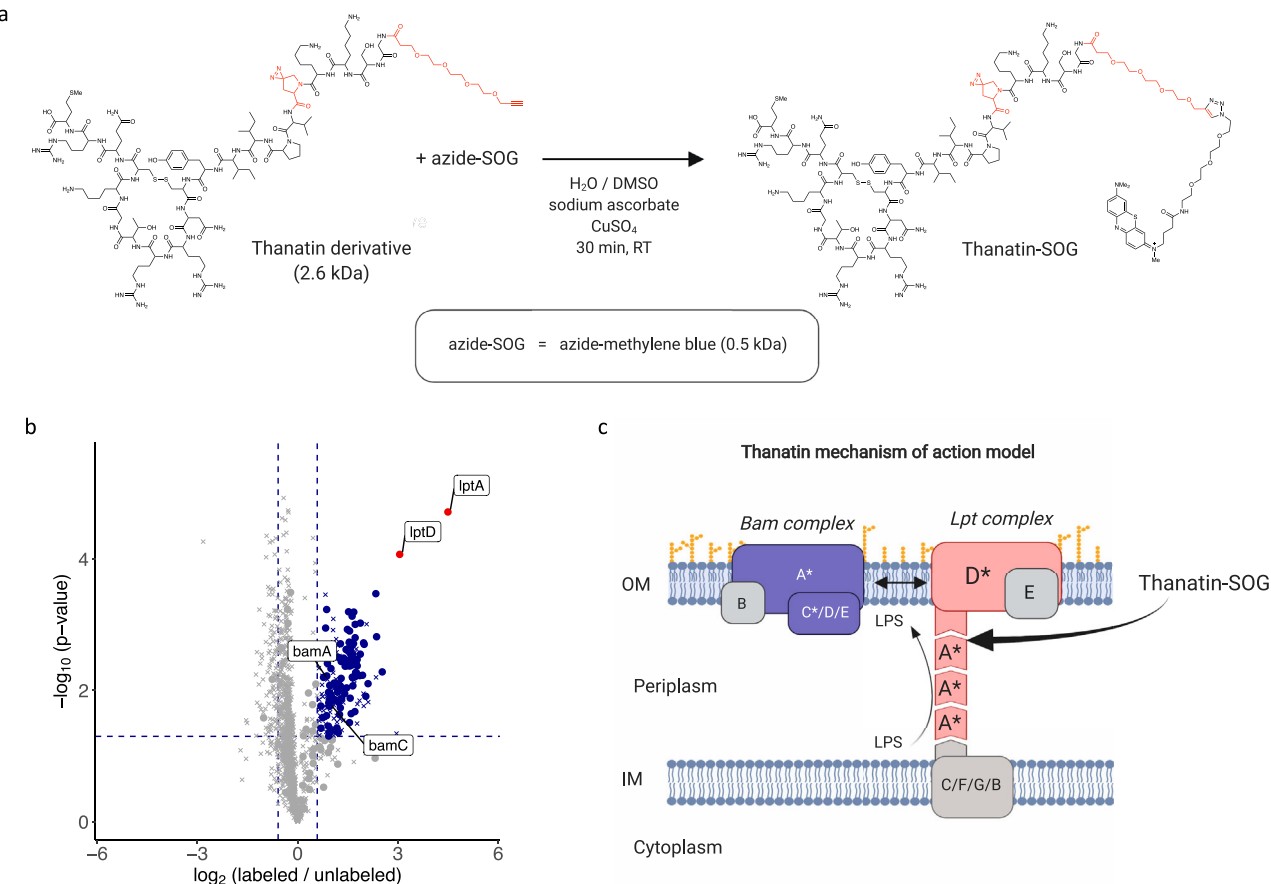

**Fig. 4 Hypothesis-free identification of surfaceome signaling interactions in prokaryotic systems. a** Schematic of coupling the peptidomimetic antibiotic Thanatin to the singlet oxygen generator (SOG) thiorhodamine. **b** Volcano plot showing relative abundance changes of LUX-MS quantified proteins from Thanatin-SOG treated *Escherichia coli* illuminated for 5 min with and without Thanatin competition, tested using a two-sided Student's *t* test. Blue dots represent significantly enriched proteins and red dots represent the direct binding targets of Thanatin previously found by photo-crosslinking experiments. The main targets and selected surfaceome interactors are highlighted. **c** Schematic representation outlining the molecular mechanism of action of Thanatin in *E. coli* cells with spatial context. Source data are provided as a Source Data file and interactive volcano plots (Supplementary Data 4).

surface expression of a putative cell wall-anchored protein and an ABC transporter that were strongly enriched by LUX-MS (Fig. 5f). Altogether, our results demonstrate the capability of LUX-MS to unravel host surfaceome interactions of intact viruses and its use to map the cell surface of pathogenic bacteria eventually facilitating the design of anti-infective strategies.

**Elucidating intercellular surfaceome signaling domains in functional immunosynapses.** Contact-dependent intercellular communication is an important regulator of immune system function and relies on highly complex intercellular surfaceome signaling architectures - in the case of T lymphocyte activation by antigen-presenting cells (APCs)—the immunological synapse[7]. The contextual interplay of immunosynaptic receptor interactions between APCs and CD8[+] T cells leads to T-cell activation, proliferation, and differentiation enabling these cells to exert effector functions against infected, tumorigenic, or foreign cells[69]. Thus, the ability to resolve functional immunosynapses at the molecular level with spatiotemporal specificity would enable discoveries in the field of immunology and immuno-oncology. We thus coupled thiorhodamine SOG to the lymphocytic choriomeningitis virus (LCMV)-derived peptide gp33-41 (gp33, sequence: KAVYN-FATM) (Fig. 6a), an immunodominant epitope of the LCMV glycoprotein (GP) that is presented by MHC class I H-2D[b] and H-2K[b][70] and that is recognized by T-cell receptor (TCR) transgenic CD8[+] T cells of P14 mice[71]. In a two-cell system, we

showed that SOG-coupled gp33 is still presented by mouse dendritic cells (DCs) and is able to activate freshly isolated P14 CD8[+] T cells with comparable efficacy than its unconjugated counterpart (Supplementary Fig. 7). Applying light led to cell surface biotinylation of both cell types, validating the photocatalytic function of SOG-coupled gp33 and our ability to efficiently perform trans-cellular surfaceome labeling during immunosynaptic cell-to-cell contacts (Supplementary Fig. 8). We then implemented stable isotope labeling by amino acids in cell culture (SILAC) for isotopic barcoding of interacting DCs and T cells (TC) and performed LUX-MS in data-independent acquisition (DIA) mode after 30 min of co-culture to analyze mature immunological synapses (Fig. 6b). Overall, we identified 4811 protein groups that could clearly be assigned to TC (e.g., TCR, CD3, and CD8), DC (e.g., MHC class I H-2D[b] and H-2K[b]), or both based on observed SILAC-ratios (Fig. 6c). We observed light-dependent enrichment of numerous cell surface proteins including components of the primary signaling axis such as MHC class I on DC and the gp33-specific TCR as well as CD8 on TC (Fig. 6d-e, Supplementary Data 6 and 7). Furthermore, we found co-stimulatory signaling components essential for early T-cell priming such as CD86 of the CD86-CD28 axis and the entire CD2-CD48 axis with reported cellular directionality[7]. Our results further indicate immunosynaptic localization of CD5 and CD6 on TC. Indeed CD5 was shown to interact with TCR, CD8, and CD2 in the immune synapse to fine-tune TCR-mediated signaling[72],

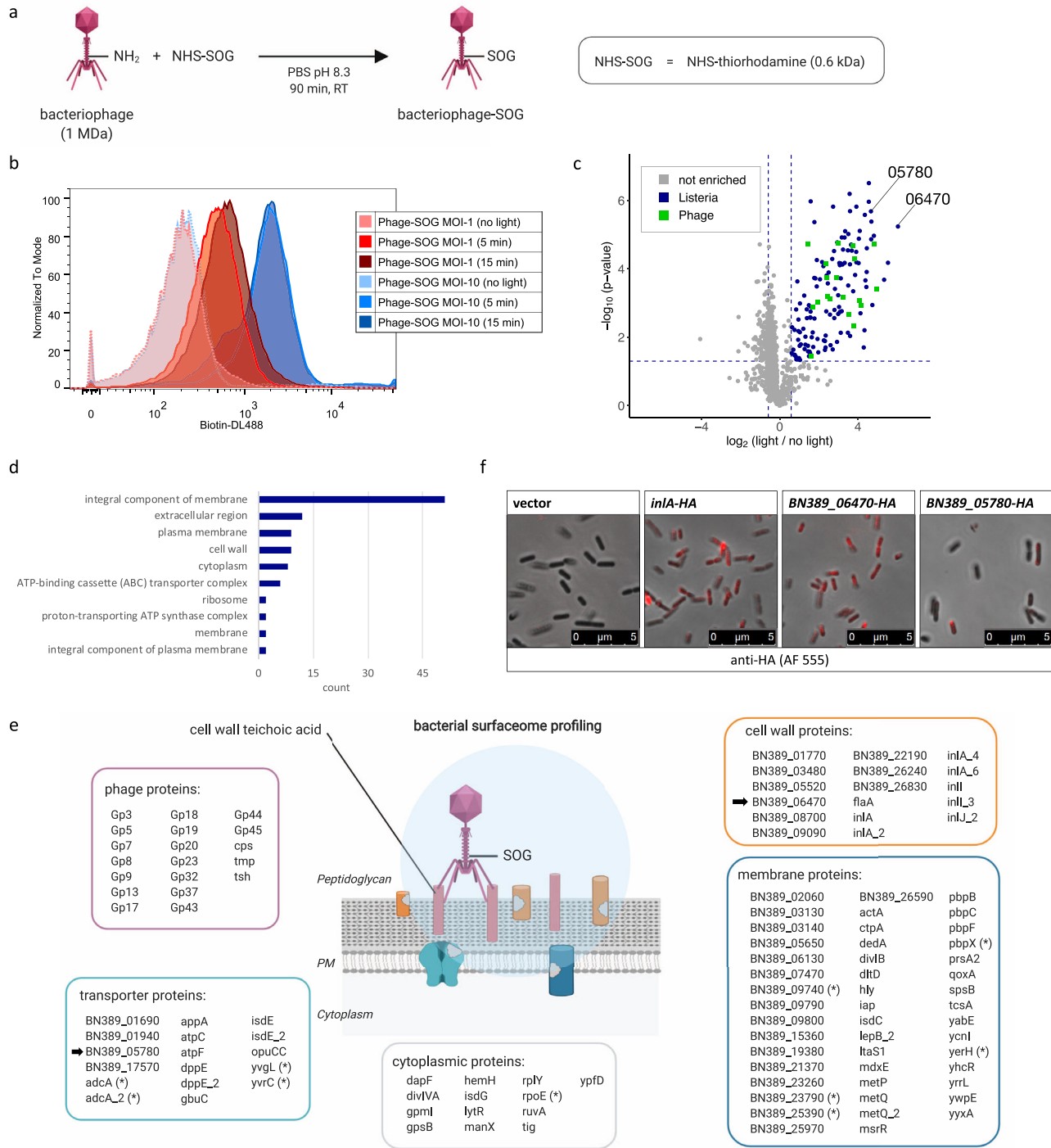

**Fig. 5 Mapping virus-targeted surfaceomes on living hosts using bacteriophage-guided LUX-MS. a** Schematic of coupling the *Listeria monocytogenes* (*Lm*)-specific bacteriophage to the singlet oxygen generator (SOG) thiorhodamine. **b** Histogram plot showing light-dependent cell surface biotinylation of *Lm* cells after LUX-labeling with bacteriophage-SOG (>30,000 cells per condition). **c** Volcano plot showing relative abundance changes of LUX-MS quantified proteins from bacteriophage-SOG treated *Lm* with and without illumination for 15 min, tested using a two-sided Student's *t* test. Blue and green dots represent significantly enriched *Lm*- or phage-derived proteins, respectively, whereas gray dots indicate non-enriched proteins. Candidates selected for microscopic characterization are highlighted. **d** GO-term enrichment analysis of significantly enriched *Lm* proteins. **e** Schematic representation of the spatial coverage of LUX-MS based on subcellular annotations of significantly enriched proteins. Candidates selected for microscopy are marked with arrows. **f** Immunofluorescence of the indicated *Lm* strains stained with an anti-HA monoclonal antibody conjugated with Alexa Fluor 555 (Red). Images are representative of three experiments. Source data are provided as a Source Data file and interactive volcano plots (Supplementary Data 5).

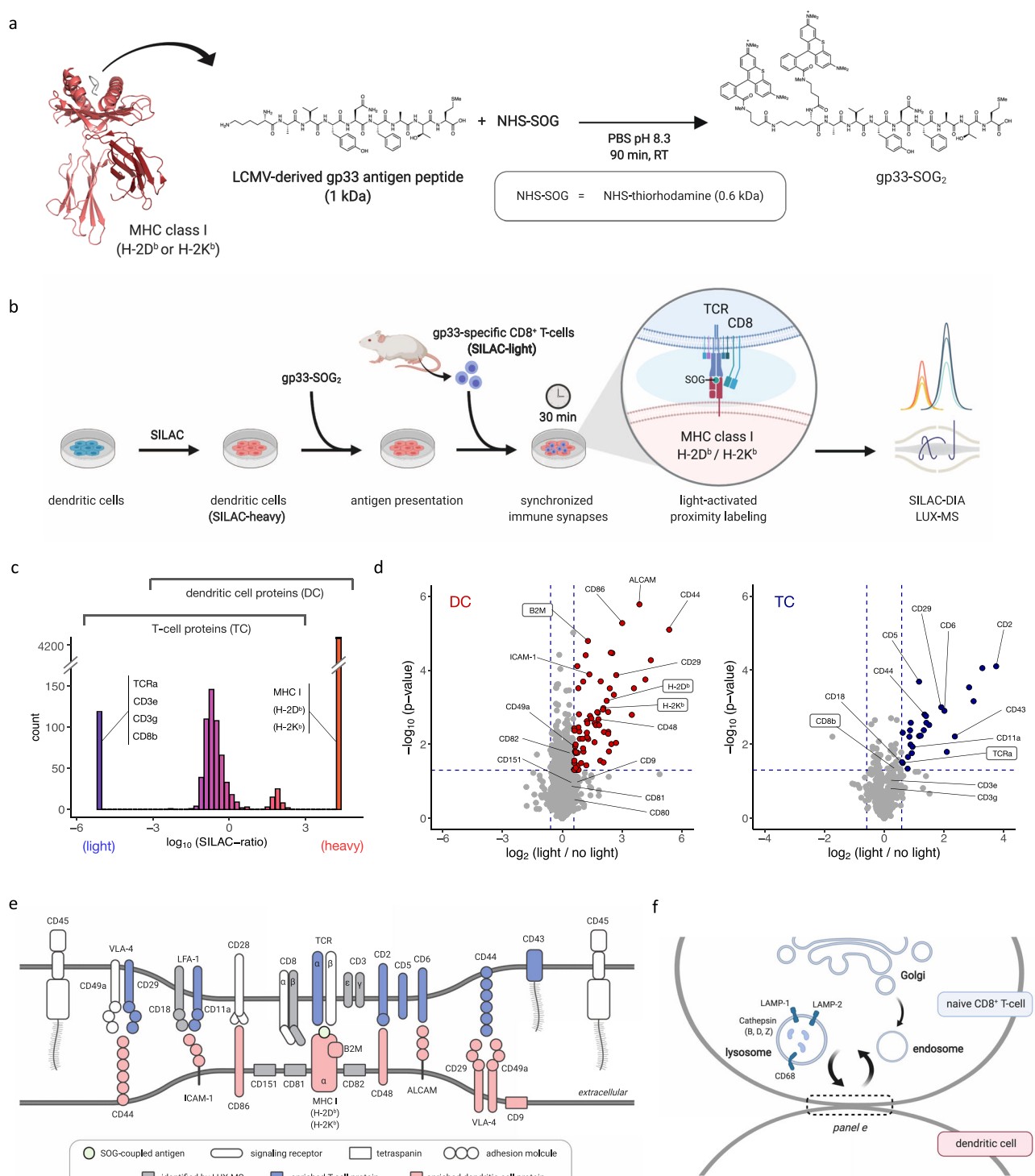

while the structurally related CD6 physically associates with CD5 and its ligand CD166 (also known as ALCAM)[73] that was also captured on DC. LUX-MS further revealed immunosynaptic adhesion complexes including LFA-1 (TC), ICAM-1 (DC), and VLA-4 (both) that enhance intercellular proximity and T-cell activation[69] and identified highly membrane-embedded tetra-spanins CD9, CD81, CD82, and CD151 on DC that organize immunosynaptic signaling domains on APCs[74]. In contrast, the ubiquitous protein tyrosine phosphatase receptor type C (PTPRC, CD45) was not identified on TC in line with its immunosynaptic exclusion to permit TCR phosphorylation and signaling[69] (Fig. 6e). Taken together, our results demonstrate the utility of

LUX-MS to comprehensively resolve the intercellular surfaceome signaling architecture of functional immune synapses with cell-type specificity and spatiotemporal control by light, providing a molecular framework for drug discovery in immunology and immuno-oncology.

## Discussion

The nanoscale organization and extracellular interaction network of the cellular surfaceome is a key mediator in antibody signaling, drug action, hormone function, viral infection, and intercellular communication. In this study, we developed and applied a

**Fig. 6 Elucidating intercellular surfaceome signaling domains in functional immunosynapses. a** Schematic of coupling immunogenic peptide gp33 to the singlet oxygen generator (SOG) thiorhodamine. The crystal structure of gp33 (dark gray) presented by MHC class I H-2D$^B$ alpha chain (red) in complex with beta-2 microglobulin (dark red) is shown (PDB identifier: 1FG2), **b** Schematic of the SILAC-DIA based LUX-MS workflow. Isotopically labeled (heavy) mouse dendritic cells presenting gp33-SOG2 establish synchronized immunosynapses with isolated (light), gp33-specific CD8$^+$ T cells enabling light-activated in situ labeling and molecular analysis of intercellular surfaceome signaling interactions within synapses using data-independent acquisition (DIA) mass spectrometry. **c** Histogram plot showing distribution of heavy-to-light abundance ratios of LUX-MS identified proteins. Ratio-boundaries for cell type assignment and representative proteins are shown. **d** Volcano plot showing relative abundance changes of LUX-MS quantified proteins from the two-cell systems with and without illumination for 15 min, tested using a two-sided Student's $t$ test. Dots and crosses represent cell surface and otherwise annotated proteins, respectively. Red and blue dots represent dendritic cell (DC) and T-cell (TC)-derived proteins, respectively. Known immunosynaptic constituents are highlighted with direct gp33 interactors shown in boxes. **e** Schematic of the LUX-MS identified immunosynaptic surfaceome interaction network.
**f** Schematic highlighting the interconnectivity of the immunological synapse with endolysosomal pathways in T cells and showing proteins reported at the cell-to-cell interface by LUX-MS. Source data are provided as a Source Data file and interactive volcano plots (Supplementary Data 6 and 7).

SOG-based and highly versatile proximity detection technology termed LUX-MS that enables to specifically resolve surfaceome signaling domains in all these scenarios with spatiotemporal control by light. In contrast to larger enzymes (APEX2: 28 kDa; miniTurbo: 27 kDa)[75], the use of small molecular SOG (<1 kDa) as labeling probes offers the following advantages: (i) Broad ligand-compatibility enabling target deconvolution and surfaceome interaction profiling of ligands of basically any description ranging from small molecules, antibodies and viral particles to whole cells. (ii) Covalent SOG-coupling with ligand-compatible chemistry preserving structure-function relationships and circumventing the need for genetic manipulation. (iii) Light-controlled generation of SO species with modifiable reactivity allowing for fine-tunable in situ labeling with tight spatiotemporal control. (iv) Protein labeling via conserved amino acid residues enables proteome-wide surfaceome interrogations across domains of life. Recently, photocatalysts producing short-lived carbene with nanosecond half-life were utilized for antibody-guided mapping of protein microdomains that could not be resolved using enzyme-based strategies (intermediates with half-lives of 0.1 ms to 60 s)[28]. Proximity labeling by LUX-MS is mediated by photosensitized SO species with a lifetime of 3.5 μs[36] that is tunable by modulating buffer conditions indicating spatial resolution in between enzyme- and carbene-based approaches. Indeed, although CD20 microenvironment mapping using LUX-MS provided a defined list of proximity candidates that was extendable by the use of D$_2$O, an HRP-based approach provided a substantially longer list of complementary CD20 candidates. Furthermore, the direct coupling of SOG to signaling competent ligands confines labeling to signaling relevant surfaceome domains and circumvents the use of antibodies that provide structural flexibility. Thus, LUX-MS complements currently available technologies and provides an analytical stepping stone for system-wide elucidation of the cellular surfaceome landscape.

Using antibodies to guide proximity labeling, we demonstrated the ability of LUX-MS to specifically reveal distinct surfaceome proximities on living cells that form a molecular framework for the reconstruction of the surfaceome nanoscale organization. Given the broad availability of high-quality antibodies for established surface targets and the high-throughput capability of the LUX-MS workflow, profound surfaceome interaction mapping is feasible providing the cell surface level information that is currently lacking in ongoing systems biology efforts to map and understand the structure and function of the human interactome[76–79].

By applying LUX-MS to small molecules, we mapped the surfaceome signaling domains of the small-molecule drug cardiac glycoside CG1. In contrast to LRC methodologies, LUX-MS not only enabled proteome-wide target deconvolution of small molecules but also provided cell-type-specific surfaceome proximity information that could be leveraged for targeted drug delivery, e.g., using EDC[12]. While this approach is expected to be transferable to other drug modalities, the accumulating knowledge of drug-targeted cellular surfaceomes eventually allows for the identification of disease-associated receptor proximites enabling the rational design of next-generation precision medicines that go well beyond the targeting of individual receptors.

Using LUX-MS, we decoded viral surfaceome interactions in bacterial hosts that are not detectable using state-of-the-art technologies such as CSC, LRC, and HATRIC owing to the structural complexity of the cell envelope and a general lack of protein glycosylation. Using bacteriophages that recognize a ubiquitous cell wall component of *Listeria monocytogenes*, we mapped the bacterial surfaceome (>80 proteins) with substantial coverage and specificity that superseded common profiling strategies such as trypsin shaving (11 proteins identified), surface-restricted biotinylation (27 proteins identified), and cell fractionation (21 proteins identified)[80]. Depending on the binding specificity of the employed viral particle, LUX-MS enables the spatiotemporal elucidation of viral attachment sites, target receptors, and global surfaceomes of live prokaryotic and eukaryotic cells in complex biological systems. This is of particular interest considering the broad application of advanced phage display assays[81] and related screening platforms[82,83] in basic and translational research.

Finally, we employed LUX-MS for the spatiotemporal analysis of intercellular receptor interaction networks within immunological synapses. Previous attempts were limited to molecular analysis in model systems that use soluble, plate-bound, or membrane-associated factors to mimic immunosynaptic T-cell activation[69] or to fluorescence-based reporting of transient intercellular immune cell interactions[28,84]. Here, we performed a gp33 immunogen-guided LUX-MS in an isotopically barcoded two-cell system of millions of synchronized and functional immunosynapses and thereby obtained a holistic view of the mature immunosynaptic surfaceome signaling architecture, fostering the formulation of original biological hypotheses. For example, we found several lysosomal proteins (LAMP-1, LAMP-2, CD68, and cathepsins B, D, and Z) within the immunosynaptic cleft on naive CD8$^+$ T cells but not antigen-presenting DCs. This indicates local fusion of lysosomal granules in T cells that is typically observed in cytotoxic T cells during degranulation-mediated target cell killing[85]. However, while being a feature of antigen-experienced cells, degranulation and thus cytolytic activity was shown to be absent in naive CD8$^+$ T cells. Indeed, we did not identify any cytolytic effector molecules such as perforin or granzymes, indicating the secretion of non-lytic granules in naive CD8$^+$ T cells within 30 min of initial contact with APCs (Fig. 6f). Albeit, the cellular machinery for immunosynapse-directed transport of vesicles is in place within 6 min of initial cell contact, the exact secretion mechanism and functional role of such non-lytic granule secretion in antigen-dependent T-cell priming remains to be further explored.

In conclusion, we provide a high-throughput, optoproteomic technology termed LUX-MS to discover protein networks within specific surfaceome signaling domains on living cells and we demonstrate its utility and versatility in a broad range of biomedically important scenarios including antibody signaling, drug action, hormone function, viral infection and intercellular communication. Given the biological and therapeutic importance of the surfaceome signaling landscape, we envision LUX-MS to impact basic and applied research areas by unraveling the spatiotemporal interconnectivity of the cell surface signaling gateway in health and disease.

## Methods

**Reagents.** All chemicals were purchased from Merck unless stated otherwise. SOGs in form of NHS-functionalized thiorhodamine (cat: AD-Thio12-41) and azide-functionalized methylene blue (cat: AD-MB2-31) were purchased from ATTO-TEC GmbH (Siegen, Germany). Biocytin-hydrazide was purchased from Pitsch Nucleic Acids AG (Stein am Rhein, Switzerland). The following antibodies were used for LUX-MS experiments: mouse IgG Isotype control antibody (Invitrogen, cat: 10400 C), anti-human CD20 mouse IgG2 monoclonal antibody clone 2H7 (Invitrogen, cat: 14-0209-82), anti-human CD38 chimeric monoclonal antibody kindly provided by Centrose LLC, anti-human CD54 mouse IgG1 monoclonal antibody, clone HCD54 (BioLegend, cat: 322704), anti-human CD166 mouse IgG1 monoclonal antibody clone 3A6 (BioLegend, cat: 343902), anti-human CD220 mouse IgG2 monoclonal antibody, clone B6.220 (BioLegend, cat: 352602).

**Cell culture.** All chemicals for cell culture were purchased from ThermoFisher Scientific unless stated otherwise. All other chemicals were purchased from Sigma-Aldrich unless stated otherwise. Cell lines were purchased from ATCC and grown at 37 °C and 5% ambient $CO_2$. Patient-derived B-lymphoma cell line SUDHL6 (ATCC, CRL-2959) and human Burkitt lymphoma B cell-line Ramos (ATCC, CRL-1596, kindly provided by Michael Reth) were grown in Roswell Park Memorial Institute (RPMI) 1640 medium with 1.5 mM GlutaMAX, 1% penicillin–streptomycin, and 10% fetal bovine serum. Patient-derived promyelocytic leukemia cell line HL60 (ATCC CCL-240) was grown in RPMI with 1.5 mM GlutaMAX (Gibco), 1% penicillin–streptomycin (Gibco), and 10% human serum (Chemie Brunschwig). Mouse dendritic cell line MutuDC1940[86] was kindly provided by Hans Acha-Orbea (Department of Biochemistry, University of Lausanne, Switzerland) and was cultured and isotopically labeled in Iscove's Modified Dulbecco's Medium for SILAC (Thermo, cat: 88367) supplemented with glucose (final: 4.5 g/l), L-Arg-$^{13}C_6$-$^{15}N_4$ (final: 42 µg/ml), L-Lys-$^{13}C_6$-$^{15}N_2$ (final: 73 µg/ml), 1.5 mM GlutaMAX, 10 mM HEPES, 50 µM β-mercaptoethanol, 1% penicillin–streptomycin, non-essential amino acids, and 10% dialyzed fetal bovine serum.

**Mice.** Female or male mice of six to sixteen weeks of age were used for animal experiments described in this study. CD45.1 P14 mice with T-cell receptors on CD8$^+$ T cells specific for the glycoprotein GP33-41 epitope of LCMV[71] crossed with Nur77-GFP mice[87] (P14 Nur77-GFP) were bred at the ETH Phenomics Center at a light–dark cycle of 12 h, the ambient temperature of 22 °C ± 2 °C and humidity of 55% ± 10%. All animals were bred and held under specific-pathogen-free conditions prior to use. All animal experiments were performed in accordance with institutional policies and Swiss federal regulations, following guidelines and being approved by the veterinary office of the Canton of Zürich (animal experimental permissions: 115/2017).

**Light sources.** Photo-oxidation reactions were controlled using Precision LED spotlights operated via a BioLED Light Control Module (BLS-PL04-US) in continuous wave mode (Mightex Systems, Pleasanton, USA). For LUX-MS applications with thiorhodamine as SOG, spotlights BLS-PLS-0590-030-05-S were used for illumination at 590 nm with a light intensity of 4.6 mW/cm². For LUX-MS applications with methylene blue as SOG, spotlights BLS-PLS-0656-030-07-S were used for illumination at 656 nm with a light intensity of 14.9 mW/cm².

**Light-controlled production of SO.** The SOG thiorhodamine was mixed with 1 µM Singlet Oxygen Sensor Green (SOSG, Invitrogen, cat: S36002) in phosphate-buffered saline at increasing $D_2O/H_2O$ ratios and illuminated at a working distance of 30 cm with 590-nm Precision LED spotlights for 0–15 min. The fluorescence of photo-oxidized SOSG was monitored using a Synergy HT Multi-Mode Plate Reader (BioTek) at excitation 485 ± 20 nm and emission 528 ± 20 nm. Quantified intensities were blanked and normalized to non-illuminated control using a customized R-script.

**Discovery and tuning of light-induced protein labeling sites.** Human holo-transferrin (Sigma, cat: T4132) was coupled with thiorhodamine SOG at a molar ligand-to-SOG ratio of 1:1 and purified using 7-kDa ZebaSpin Columns (Thermo,

cat: 89882) according to manufacturer's recommendations. SOG-coupled transferrin in $D_2O$-based phosphate-buffered saline (PBS) containing 0 or 5 mM biocytin-hydrazide was either illuminated with 590-nm Precision LED spotlights for 15 min or left in the dark. The buffer was complemented with 50 mM 2-(aminomethyl)imidazole dihydrochloride catalyst for 50 min to catalyze hydrazone formation. Treated proteins were digested overnight at 37 °C using sequencing-grade trypsin (Promega, cat: V511C) at an enzyme-to-protein ratio of 1:100. Peptides were C18-purified using 5–60 µg UltraMicroSpin Columns (The Nest Group, cat: SEMSS18V) according to manufacturer's instructions and subjected for mass spectrometric analysis using an Orbitrap Fusion Tribrid mass spectrometer (Thermo Scientific) in data-dependent acquisition (DDA) mode with a mass resolution of 120,000 and 30,000 on precursor and fragment level, respectively (see below for details). Acquired raw files of illuminated samples were subjected to an open modification search using MSfragger (v.20190628)[88] and Crystal-C[89] within the FragPipe pipeline (v.9.4) using standard settings and a SwissProt-reviewed human protein database (downloaded June 2014) containing common contaminants. Observed modifications were implemented in a subsequent closed search of all acquired samples using Comet (v.2015.01) within the Trans Proteomic Pipeline v.4.7 (SPC/ISB Seattle) as variable modification (carbonylation of His, Cys, and Trp; single oxidation of Met, Trp, and Tyr; double oxidation of Met, Trp, and His; triple oxidation of Met, Cys, and Trp) to identify fully tryptic peptides with a maximum of two missed cleavage sites and a maximum of five variable modifications. The precursor and fragment mass tolerance was thereby set to 20 ppm and 0.02 Da, respectively. Peptides were quantified by integration of chromatographic traces using Progenesis QI v.4.0 (Nonlinear Dynamics) and filtered to a false discovery rate of <1%. Transferrin peptides were assigned to dummy proteins representing unmodified or single modification types. Within the R computing environment (v.3.4.0), protein abundance fold changes (expressed in log2) were calculated using a linear mixed-effect model and tested for statistical significance using a two-sided $t$ test with the appropriate degree of freedom in the R package MSstats (v.3.8.6)[90]. Modification types with an abundance fold change >1.5 upon illumination and an adjusted $p$ value < 0.05 were considered products of light-induced photo-oxidation. Analogously, modification types with a significantly reduced light-dependent production in the presence of biotin-hydrazide ($p$ value < 0.1) were considered hydrazide-reactive amino acid modifications. In total, three biologically independent replicates were used per condition.

For amino acid-specific tuning of protein labeling sites by varying buffer conditions, human holo-transferrin (Sigma, cat: T4132) was digested overnight at 37 °C using sequencing-grade trypsin (Promega, cat: V511C) at an enzyme-to-protein ratio of 1:50. Peptides were desalted using Waters Sep-Pak C18 Column 50 mg (Waters, cat: WAT054955) according to the manufacturer's instructions and dried in a SpeedVac concentrator (Thermo Scientific). Peptides were resuspended in either PBS, 50% $D_2O$-based PBS or 100% $D_2O$-based PBS to a final peptide concentration of 10 µM and then supplemented with Tris quenched thiorhodamine SOG at a molar peptide-to-SOG ratio of 1:1. Samples were either illuminated with 590-nm Precision LED spotlights for 15 min or left in the dark. All samples were complemented with PBS or $D_2O$-based PBS to reach the final 50% D2O and incubated for 2 h in the dark. Peptides were C18-purified using 7–70 µg BioPureSPNmini Columns (The Nest Group, cat: HUM S18V) according to manufacturer's instructions and dried in a SpeedVac concentrator. The modified transferrin peptides were subjected for mass spectrometric analysis using an Orbitrap Q Exactive HF mass spectrometer (Thermo Scientific) in DDA mode with a mass resolution of 60,000 and 15,000 on precursor and fragment level, respectively (see below for details). Acquired raw files of illuminated samples were subjected to a closed search using SEQUEST HT and Multi Peptide Search in Proteome Discoverer (v.2.5.0.400, Thermo Scientific) and a SwissProt-reviewed human protein database (downloaded June 2014) containing common contaminants. Modifications were set as above and precursor and fragment mass tolerance were set to 10 ppm and 0.02 Da, respectively. Label-free quantification was performed on precursor level using only unique peptides. As above, transferrin peptides were assigned to dummy proteins representing unmodified or single modification types. Within the R computing environment (v.3.4.0), protein abundance fold changes (expressed in log2) were calculated using a linear mixed-effect model and tested for statistical significance using a two-sided t-test with the appropriate degree of freedom in the R package MSstats (v.3.8.6)[90] and visualized as dot plot over all conditions using R. In total, one biologically independent sample was prepared per condition and measured three times by mass spectrometry.

**Collection and purification of human PBMCs.** Buffy coats were obtained from healthy donors provided by the Blood Transfusion Service Zurich. To isolate PBMCs, human buffy coat samples were diluted 1:1 in PBS (Gibco) and mononuclear cells were isolated with a Histopaque-1077 density gradient (Sigma-Aldrich) according to the manufacturer's instructions. PBMCs at the interface were collected, washed once in PBS, and resuspended in media until further processing.

**Generation of antibody-SOG and HRP conjugates.** For antibody-SOG conjugates, 10 µg antibody per sample was purified using 7-kDa ZebaSpin Desalting Columns (Thermo Scientific, cat: 89882), following manufacturer's guidelines, and eluted in 10 mM sodium bicarbonate in PBS, pH 8.3. Antibody was coupled with

thiorhodamine SOG at a molar antibody to SOG ratio of 1:5, purified using 7-kDa ZebaSpin Columns, and immediately used for LUX-MS experiments.

For antibody-HRP conjugates, 10 μg anti-CD20 antibody (Invitrogen, clone 2H7) per sample was purified using 7-kDa ZebaSpin Desalting Columns (Thermo Scientific, cat: 89882), following manufacturer's guidelines, and eluted in 0.2 M carbonate-bicarbonate buffer, pH 9.4. Purified anti-CD20 antibody was conjugated to pre-activated amine-reactive HRP (Thermo Scientific™ EZ-Link™ Plus Activated Peroxidase, Thermo Scientific, cat: 31489) for 1.5 h in 0.2 M carbonate-bicarbonate buffer, pH 9.4 with a coupling ratio of antibody to HRP of 1:10. The reaction mixture was cleaned with 7-kDa ZebaSpin Desalting Columns, eluted in PBS, pH 7.2, and further processed according to the manufacturer's manual. Anti-CD20-HRP conjugates were immediately used for the HRP-based proximity labeling experiment.

For CG1-HRP conjugates, pre-activated amine-reactive HRP (Thermo Scientific™ EZ-Link™ Plus Activated Peroxidase, Thermo Scientific, cat: 31489) was coupled to amine-containing cardiac glycoside CG1 (kindly provided by Centrose LLC, Madison, Wisconsin) for 1.5 h in 0.2 M carbonate-bicarbonate buffer, pH 9.4 with a coupling ratio of HRP to CG1 of 1:10. HRP-CG1 conjugate was re-buffered into PBS, pH 7.2 using 7-kDa ZebaSpin Desalting Columns and further processed as recommended by the manufacturer's guidelines. HRP-CG1 conjugates were purified and separated from free CG1 using both 7-kDa ZebaSpin Desalting Columns (twice) and 10 kDa molecular weight cut-off filters (Merck, cat: UFC501024) and subsequently used for single-cell chemosensitivity screening.

**Degree of labeling (DOL) calculation**. Average DOL of antibodies labeled with thiorhodamine SOG probes was determined by measuring absorbance as previously described[12]. Specifically, the absorbance of conjugates (Conj), antibody (Ab), and free thiorhodamine (SOG) at both 280 nm and 582 nm were measured using a NanoDrop 2000 spectrophotometer (Thermo Scientific). Next, the following ratios were calculated; $Ratio_{Ab} = A582_{Ab}/A280_{Ab}$ and $Ratio_{SOG} = A582_{SOG}/A280_{SOG}$. Then, the absorbance of the antibody-SOG conjugate was measured yielding $A280_{Conj}$ and $A582_{Conj}$. The actual concentration of antibody and SOG contained in the conjugate solution were determined as follows: $[Ab] = (A280_{Conj} - (A582_{Conj} - Ratio_{Ab} \times A280_{Conj})/(Ratio_{SOG} - Ratio_{Ab}))/\varepsilon_{Ab}$, $[SOG] = (A582_{Conj} - Ratio_{Ab} \times A280_{Ab})/\varepsilon_{SOG}$. The following extinction coefficients ($\varepsilon$) were thereby used: $\varepsilon_{Ab} = 210,000\ M^{-1}\ cm^{-1}$, $\varepsilon_{SOG} = 110,000\ M^{-1}\ cm^{-1}$. Finally, the DOL was determined: $DOL = [SOG]/[Ab]$. Per conjugate, an average DOL of six replicate measurements was calculated in the R computing environment. The same procedure was applied for measuring the DOL of HRP labeled with CG1 by measuring absorbance at both 403 nm and 299 nm and by using $\varepsilon_{HRP} = 100,000\ M^{-1}\ cm^{-1}$ and $\varepsilon_{CG1} = 5623\ M^{-1}\ cm^{-1}$.

**Antibody-guided receptor microenvironment mapping**. For SOG-based LUX-MS, $20 \times 10^6$ cells were incubated with 10 μg SOG-coupled antibody (final 5 μg/ml) for 30 min at 4 °C in the dark to minimize background light-induced oxidation. Cells were washed with ice-cold PBS, resuspended in chilled photo-oxidation buffer (5 mM biocytin-hydrazide in D$_2$O-based PBS, pH 7.5 if not stated otherwise), and illuminated at a working distance of 20 cm with 590-nm Precision LED spotlights or left in the dark. For the anti-CD20 time course LUX-MS, one and three biologically independent samples were analyzed for each illumination time point and the non-illuminated control, respectively. For anti-CD20 LUX-MS two and three biologically independent samples were analyzed for labeled and unlabeled conditions, respectively. For all other antibody-guided LUX-MS, three biologically independent samples were analyzed for both labeled and unlabeled conditions. Cells were pelleted by centrifugation and resuspended in chilled labeling buffer (5 mM biocytin-hydrazide and 50 mM 2-(Aminomethyl)imidazole dihydrochloride in PBS, pH 6.0) for 50 min at 4 °C in the dark. For flow cytometric analysis of cell surface biotinylation, cells were extensively washed with PBS and stained with NeutrAvidin Protein DyLight 488 (Invitrogen, cat: 22832) 1:200 for 20 min prior analysis using an Accuri C6 Flow Cytometer (BD Biosciences) and FlowJo (v.10.07). The FSC/SSC pattern of the unlabeled condition was thereby used to gate for live cells (Supplementary Fig. 12). For LUX-MS analysis, cells were extensively washed with PBS, snap-frozen as cell pellets in liquid nitrogen, and stored at −80 °C until further processing.

For HRP enzyme-based proximity labeling, $20 \times 10^6$ Ramos cells were harvested per sample and twice washed with ice-cold PBS. Cells of all conditions were incubated with CD20-HRP-conjugate in 1 ml PBS for 30 min at 4 °C. After incubation, cells were washed once with ice-cold PBS and once with room temperature (RT) PBS. Afterwards, cells were resuspended at RT in PBS containing 0.1 mM biotin-tyramide (Pitsch Nucleic Acids AG). The HRP-based proximity labeling reaction was started by the addition of H$_2$O$_2$ (final 1 mM) to the CD20 labeling condition. For the control samples, PBS-based Mock buffer (without H$_2$O$_2$) was used. After 1 min, the reaction was stopped with a quenching buffer containing 10 mM NaN$_3$, 10 mM sodium ascorbate, and 1:10,000 catalase (Sigma-Aldrich, C30) in ice-cold PBS, immediately followed by three washes with quenching buffer and two washes with ice-cold PBS. Cells were snap-frozen as cell pellets in liquid nitrogen and stored at −80 °C until further processing. In total, three biologically independent replicates were used per condition.

**Synthesis of SOG-coupled cardiac glycoside CG1**. The primary amine-containing cardiac glycoside CG1 was kindly provided by Centrose LLC (Madison, Wisconsin). To a solution of CG1 **2** (5 mg, 0.0081 mmol) and thiorhodamine-NHS ester **1** (5.7 mg, 0.0081 mmol) in N,N-dimethylformamide (1 mL) was added Et$_3$N (0.011 mL, 0.08 mmol). The reaction mixture was stirred at RT for 1 h and product formation was confirmed by LC-MS. The final product CG1-SOG **3** was purified by preparative RP-HPLC on a Waters Eclipse XDB (C18, 250 × 21.2 mm, 7 μm) column with a gradient of 5 –50% ACN/H$_2$O + 0.1% TFA in five column volumes. The product was of >95% purity as judged by reversed-phase UPLC and HR-MS. HR-ESI-MS: m/z (M + H$^+$) 999.4938 (calc. mass = 999.4936) (Supplementary Fig. 9).

**Single-cell chemosensitivity screening**. A single-cell suspension of HL60 cells was seeded (2000 cells/well with 50 μl/well) in CellCarrier 384 Ultra, clear-bottom, tissue-culture-treated plates (PerkinElmer) containing antineoplastic agents (see below) and incubated overnight for 24 h at 37 °C and 5% CO$_2$. Cell number and viability were determined by the use of a Countess II Cell Counter (Thermo Fisher). Cells were subsequently treated with a drug screen library of three single compounds (CG1, CG1-SOG, and CG1-HRP) in four concentrations each with ten technical replicates that were dispensed into assay plates by an Echo liquid handler (Labcyte). The assay was stopped by fixing and permeabilizing the cells with 20 μl/well of a solution containing 0.5% (w/v) formaldehyde, 0.05 % (v/v) Triton X-100, 10 mM sodium(-meta)periodate, and 75 mM L-lysine monohydrochloride, after removing the culturing media. After 20 min incubation at RT, the fixative-containing media was aspirated by use of a HydroSpeed plate washer (Tecan). The cells were then blocked (50 μl/well) with PBS supplemented with 5% fetal bovine serum (FBS, Gibco) and stored until immunostaining. For nuclear detection, 2 μg/mL DAPI (4′,6-Diamidino-2-Phenylindole, Biolegend) in PBS was utilized. Cells were further stained with an Alexa Fluor 647 anti-human CD33 antibody (Clone P67.6) at a dilution of 1:20. Before the staining, the blocking solution was removed, 20 μl/well of the antibody/DAPI cocktail was added and the plate was incubated for 1 h in the dark (at RT). Subsequently, the staining solution was removed and PBS was added (70 μL/well) before imaging. The 384-well plate was imaged with an Opera Phenix automated spinning-disk confocal microscope (PerkinElmer) at ×20 magnification with 5 × 5 non-overlapping images, covering the whole well surface. Each well of the plate was fully imaged in the brightfield (650–760 nm), DAPI/Nuclear signal (435–480 nm), and APC/Red signal (650–760 nm) channels. Raw.tiff images were used for single-cell image analysis in CellProfiler (v.2)[91]. Single cells detection and nuclear segmentation were performed by maximum correlation thresholding on the DAPI channel. To extract cytoplasmic measurement, cellular outlines were estimated by a circular expansion of 10 pixels around the nucleus. To measure the local intensity background around every single cell, a second larger expansion of 30 pixels was performed. Standard CellProfiler raw fluorescent intensities were extracted, log10 transformed, and normalized towards the local cellular background[92].

**Identification of small-molecule drug-targeted surfaceome structures**. Fifty million cells were incubated with 0–10 μM CG1-SOG (10 μM for LUX-MS experiment) for 30 min at 4 °C in the dark to minimize background light-induced oxidation. Cells were washed with ice-cold PBS, resuspended in chilled photo-oxidation buffer, and illuminated at a working distance of 20 cm with 590-nm Precision LED spotlights or left in the dark (three biologically independent samples each). Cells were pelleted by centrifugation and resuspended in chilled labeling buffer for 50 min at 4 °C in the dark. For flow cytometric analysis of cell surface biotinylation, cells were extensively washed with PBS and stained with NeutrAvidin Protein DyLight 488 (Invitrogen, cat: 22832) 1:200 for 20 min prior analysis using an Accuri C6 Flow Cytometer (BD Biosciences) and FlowJo (v.10.07). The FSC/SSC pattern of the unlabeled condition was thereby used to gate for live cells (Supplementary Fig. 12). For LUX-MS analysis, cells were extensively washed with PBS, snap-frozen as cell pellets in liquid nitrogen, and stored at −80 °C until further processing.

**Biomolecule-guided discovery of intercellular surfaceome signaling domains**. The human peptide hormone insulin and glycoprotein holo-transferrin were coupled with thiorhodamine SOG at a molar ligand:SOG ratio of 1:1 and 1:13, respectively according to the manufacturer's recommendations. The SOG-coupled ligand constructs were kept at 4 °C and immediately used. Twenty million cells were incubated with 1 μM Insulin-SOG or 75 nM Transferrin-SOG for 10 min at 4 °C in the dark to minimize background light-induced oxidation. Cells were washed with ice-cold PBS, resuspended in chilled photo-oxidation buffer, and illuminated at a working distance of 20 cm with 590-nm Precision LED spotlights for 5 min or left in the dark (three biologically independent samples each). Cells were pelleted by centrifugation and resuspended in chilled labeling buffer for 50 min at 4 °C in the dark. Cells were extensively washed with PBS, snap-frozen as cell pellets in liquid nitrogen, and stored at −80 °C until further processing.

**Synthesis and characterization of SOG-coupled antimicrobial Thanatin**. A Thanatin derivative was kindly provided by the John Robinson group (University of Zurich, Switzerland) and (6.0 mg, 0.0023 mmol) were dissolved in 500 μl H$_2$O. Sodium ascorbate (0.1 M, 30 μl, 0.0022 mmol) and azide-functionalized methylene

blue as SOG (0.5 mg, 0.0009 mmol) in 100 μl DMSO and CuSO$_4$ (0.1 M, 30 μl, 0.003 mmol) was added. After 15 min the reaction mixture was directly injected into a prep HPLC C18 column with a gradient of 10–40% ACN/H$_2$O (0.1% TFA). The purity of the Thanatin-SOG compound was confirmed by UPLC analysis and the identity was verified by HR-ESI with a calculated mass of 652.1353 (M + 6H)$^{6+}$ and a measured m/z value of 652.1360 (M + 6H)$^{6+}$ (Supplementary Fig. 10).

**Proteome-wide and light-activated target deconvolution in prokaryotic systems**. *E. coli* cells (Migula) Castellani and Chalmers (ATCC: 25922) were incubated with 3 μM Thanatin-SOG in the presence or absence of 30 μM unconjugated Thanatin derivative for 30 min at 37 °C in the dark to minimize background light-induced oxidation. Cells were washed with ice-cold PBS, resuspended in chilled photo-oxidation buffer, and illuminated at a working distance of 20 cm with 659-nm Precision LED spotlights for 5 min or left in the dark (three biologically independent samples each). Cells were pelleted by centrifugation and resuspended in chilled labeling buffer for 50 min at 4 °C in the dark. Cells were extensively washed with PBS, snap-frozen as cell pellets in liquid nitrogen, and stored at −80 °C until further processing.

**Generation and characterization of SOG-coupled bacteriophages**. Gram-positive *L. monocytogenes* specific bacteriophages were coupled with thiorhodamine SOG by mixing 1.2 ml of *Listeria* A500 phage solution (a titer of 10$^{12}$ plaque-forming units (PFU)/ml) with 60 μl of 0.2 M NaHCO$_3$ and 30 μl of 0.15 mM NHS-functionalized thiorhodamine. The mixture was incubated at RT in the dark for 1 h. The phage-SOG complex was precipitated by adding 1/5 volume of 20% PEG/2.5 M NaCl and incubated for 2 h at 4 °C. The supernatant was discarded to remove unreacted SOG molecules, and the M13-dye pellet was resuspended in 1 ml of SM buffer and subjected to PFU determination. For the PFU assay, 990 μl overnight cultures of *L. monocytogenes* cells were mixed with 10 μl of serially diluted phage stock solution (in SM buffer). The culture was then mixed and vortexed with 5 ml of melted soft 1/2 BHI agar (50 °C) and poured onto the solid agar plates. Solidified agar plates were incubated at 30 °C for a day and phage titer was determined based on observed plaque numbers.

**Bacteriophage-guided exploration of virus-targeted host surfaceomes**. *Listeria monocytogenes* cells (strain WSLC 1042) were treated with 40 μg/ml gentamicin for 1 h at RT before incubation with SOG-coupled *Listeria* A500 phages at an MOI of 10 for 30 min at 4 °C in the dark. Per sample, bacteria corresponding to 3.5 × 10$^9$ colony-forming units (CFU) were washed with ice-cold PBS, resuspended in chilled photo-oxidation buffer, and illuminated at a working distance of 20 cm with 590-nm Precision LED spotlights for 15 min or left in the dark (three biologically independent samples each). Cells were pelleted by centrifugation and resuspended in chilled labeling buffer for 50 min at 4 °C in the dark. For flow cytometric analysis of cell surface biotinylation, cells were extensively washed with PBS and stained with NeutrAvidin Protein DyLight 488 1:50 for 20 min prior analysis using an Accuri C6 Flow Cytometer (BD Biosciences) with side-scatter (height) set to 10,000 as detection limit and FlowJo (v.10.07). The FSC/SSC pattern of the unlabeled condition was thereby used to gate for live cells (Supplementary Fig. 12). For LUX-MS analysis, cells were extensively washed with PBS, snap-frozen as cell pellets in liquid nitrogen, and stored at −80 °C until further processing.

**Genetic engineering and microscopic validation of identified *L. monocytogenes* surface proteins**. Positive surface protein control Internalin A (InlA) and candidate genes encoding putative surface proteins A0A0E1RAE1 (LPXTG-motif cell wall anchor domain protein) and A0A0E1R383 (ABC transporter), and their native promoters were cloned into the pPL2 integration vectors[67] with an HA tag using Gibson method as previously described[93]. The empty vector pPL2, three insert-containing vectors pPL2(inlA-HA), pPL2(rae1-HA), and pPL2(r383-HA), were transformed into electrocompetent *L. monocytogenes* 1042 cells. The primers 5′-GTCAAAACATACGCTCTTATC-3′ and 5′-ACATAATCAGTCCAAAGT AGATGC-3′ were used to verify the pPL2 integration into the bacterial genome. For surface immunostaining of *L. monocytogenes* WT and mutant strains, corresponding cells were grown to mid-log phase, washed with PBS, and then blocked in 1% BSA/PBS. Cells were further washed and stained with an HA monoclonal antibody conjugated with Alexa Fluor 555 (ThermoFisher Scientific) at a 1:100 dilution for 30 min. After final washing, cell suspensions were applied to a glass slide and imaged by confocal laser scanning microscopy.

**Synthesis and characterization of SOG-coupled immunogenic peptide gp33**. LCMV peptide gp33-41 (gp33; KAVYNFATM) was purchased from NeoMPS. Peptide gp33 (1.0 mg, 1.0 μmol, 1 equiv.) and NHS-thiorhodamine (1.4 mg, 2.0 μmol, 2 equiv.) were incubated at RT for 90 min in PBS at pH 8.3. The reaction mixture was directly purified by preparative RP-HPLC on a Jasco instrument using a YMC C18 column (5 μm, 20 mm I.D. × 250 mm) at RT at a flow rate of 10 mL/min, with simultaneous monitoring of the eluent at 220, 254, and 301 nm. Milli-Q water with 0.1% TFA (solvent A) and a gradient of 5–15% ACN with 0.1% TFA (solvent B) over 5 min and 15 to 75% solvent B over 30 min was used as the eluent. Fractions containing pure product were pooled and lyophilized to obtain gp33-SOG$_2$ as a purple solid. The purity and identity were verified by analytical RP-HPLC and HR-MS. HR-MS (ESI): calculated for C$_{106}$H$_{133}$N$_{17}$O$_{17}$S$_3^{2+}$ [M]$^{2+}$: 1005.9608, found: 1005.9589 (Supplementary Fig. 11).

**Functional assessment of SOG-coupled gp33**. CD8$^+$ T cells were isolated from smashed spleens and inguinal lymph nodes of P14 Nur77-GFP mice using the EasySep™ Mouse naive CD8$^+$ T-cell Isolation Kit (StemCell, Grenoble, France) following the manufacturer's instructions. Spleens and lymph nodes were smashed using a 70 μm cell strainer. Isotopically labeled mouse DCs were incubated with no compound or 1 μg/ml purified gp33 carrying no (gp33), one (gp33-SOG), or two (gp33-SOG$_2$) SOGs for 1 h. A 10-fold excess of acutely isolated CD8$^+$ T cells was added by centrifugation and cells were co-incubated for 3 h in the presence of interleukin-2. Cells were carefully washed with ice-cold PBS, chilled photo-oxidation buffer was added and interacting cells were illuminated at a working distance of 20 cm with 590-nm Precision LED spotlights for 15 min. Cells were pelleted by centrifugation and resuspended in chilled labeling buffer for 50 min at 4 °C in the dark. Cells were extensively washed with PBS and assessed for T-cell activation and cell-type-specific surface biotinylation using flow cytometry.

**Light-activated in situ labeling of functional immunosynapses**. Per sample, ten million isotopically labeled mouse DCs were pre-activated with 0.5 μg/ml CpG (Microsynth, Balgach, Switzerland) overnight. PBS-washed DCs were then incubated with 1 μg/ml gp33-SOG$_2$ for 1 h in the dark to minimize background light-induced oxidation. A 1.5-fold excess of freshly isolated CD8$^+$ T cells was added by centrifugation and cells were co-incubated for 30 min in the presence of interleukin-2. Cells were carefully washed with ice-cold PBS, chilled photo-oxidation buffer was added and interacting cells were illuminated at a working distance of 20 cm with 590-nm Precision LED spotlights for 15 min or left in the dark (two and three biologically independent samples, respectively). Cells were pelleted by centrifugation and resuspended in chilled labeling buffer for 50 min at 4 °C in the dark. Cells were extensively washed with PBS, snap-frozen as cell pellets in liquid nitrogen, and stored at −80 °C until further processing.

**Flow cytometry analysis of murine immune cells**. Harvested cells were stained for flow cytometry in PBS for 20 min at RT. The following antibodies were purchased from BioLegend (San Diego, United States) and used for flow cytometric analysis: CD8a-BV510 (cat: 100752, clone: 53-6.7, dilution 1:200), CD45.1-PacificBlue (cat: 110722, clone: A20, dilution 1:200), CD44-APC (cat: 559250, clone: IM7, dilution 1:100), CD11c-PerCP (cat: 117326, clone: N418, dilution 1:200), CD69-PeCy7 (cat: 104512, clone: H1.2F3, dilution 1:100), CD25-FITC (cat: 553072, clone: 3C7, dilution 1:100). The viability of cells was determined by using fixable near-IR dead cell staining (Life-Technologies, Carlsbad, United States). Biotinylation was assessed by streptavidin-PE (BioLegend) staining for 10 min at RT. T-cell activation was determined by the expression of GFP expression under the control of the Nur77 promoter. Data was acquired on a Canto™ flow cytometer (BD Bioscience, Allschwil, Switzerland) and analyzed using the FlowJo software (Treestar, Ashland, OR, USA) (see Supplementary Fig. 12 for gating strategy).

**Automated protein capture and processing**. Labeled cell pellets were lysed in 500 μl lysis buffer (100 mM Tris, 1% sodium deoxycholate, 10 mM Tris(2-carboxyethyl) phosphine, 15 mM 2-chloroacetamide) by sonication for four intervals of 30 s in a VialTweeter (Hielscher Ultrasonics) at a power of 170 W with 80% cycle time and subsequent heating to 100 °C for 5 min. The lysis buffer for bacterial cells additionally contained 75 μg Ply511 *Listeria* endolysins to diminish cell wall integrity. Protein concentration was determined using a Nanodrop 2000 Spectrophotometer (Thermo Fisher Scientific) and equal amounts between sample and control conditions (typically 3 mg protein) were subjected to automated capture and processing of photolabeled proteins using a liquid-handling robot. Specifically, in-house packed tips containing 80 μl Streptavidin Plus UltraLink resin (Thermo Fisher Scientific) were hooked to a Versette liquid-handling system (Thermo Fisher Scientific) for automated mixing with cell lysate for 2.5 h and subsequent washing with 5 M NaCl, StimLys buffer (50 mM Tris pH 7.8, 137 mM NaCl, 150 mM glycerol, 0.5 mM EDTA, 0,1% Triton X-100), 100 mM NaHCO$_3$ and 50 mM (NH$_4$)HCO$_3$. Bead-bound proteins were enzymatically digested with 0.5 μg lysyl endopeptidase Lys-C (Wako, cat no. 125-05061) for 2 h at 37 °C in 3 M urea/50 mM ammonium bicarbonate and subsequently diluted to 1.5 M urea/50 mM ammonium bicarbonate for overnight digestion with 0.8 μg sequencing-grade trypsin at 37 °C. Eluting peptides were C18-purified using 5–60 μg Ultra-MicroSpin Columns according to manufacturer's instructions and resuspended in 3% acetonitrile (ACN), 0.1% formic acid (FA) containing iRT peptides (Biognosys AG, Schlieren, Switzerland) for mass spectrometric analysis.

**Liquid chromatography-tandem mass spectrometry**. For discovery-driven identification of light-induced protein modifications, photo-oxidized transferrin peptides were resuspended in 3% ACN, 0.1% FA and separated by reversed-phase chromatography on an HPLC column (75-μm inner diameter, New Objective) that was packed in-house with a 15-cm stationary phase (ReproSil-Pur C18-AQ, 1.9 μm) and connected to a nano-flow HPLC with an autosampler (EASY-nLC 1000, Thermo Scientific). The HPLC was coupled to an Orbitrap Fusion Tribrid mass spectrometer (Thermo Scientific) equipped with a nanoelectrospray ion source (Thermo Scientific). Peptides were loaded onto the column with 100% buffer A (99.9% H$_2$O, 0.1% FA) and eluted at a constant flow rate of 300 nl/min with a 45 linear gradient from 10–32% buffer B (99.9% ACN, 0.1% FA) followed by a 4-min transition from 32 to 54% buffer B. After the gradient, the column was washed for 10 min with 98%, 4 min with 10%, and again

8 min with 98% buffer B. Samples were acquired in DDA mode with a fixed cycle time of 3 s (universal method). Electrospray voltage and capillary temperature were thereby set to 2.2 kV and 275, respectively. A high-resolution survey mass spectrum (from 395 to 1500 m/z) acquired in the Orbitrap with a resolution of 120,000 at m/z 200 (automatic gain control target value $5 \times 10^5$ and maximum injection time 100 ms) was followed by MS/MS spectra with a resolution of 30,000 of most-abundant peptide ions with a minimum intensity of $2 \times 10^4$ that were selected for subsequent higher-energy collision-induced fragmentation with a fixed collision energy of 28% and an isolation window of 2 Da. Fragments were detected by MS/MS acquisition in the orbitrap with an auto normal scan range mode, automatic gain control target value $1 \times 10^6$ and a maximum injection time 200 ms with parallelized ion injection set to on. Fragmented precursors were dynamically excluded for 10 s.

For tuning of light-induced protein modifications, photo-oxidized transferrin peptides were resuspended and separated by reversed-phase high-performance liquid chromatography as described above. Eluting peptides were eluted into a Q-Exactive HF mass spectrometer (Thermo Scientific) equipped with a nanoelectrospray ion source (Thermo Scientific) and analyzed in DDA mode. A survey mass spectrum (from 350 to 1650 m/z) acquired in the Orbitrap with a resolution of 60,000 at m/z 200 (automatic gain control target value $3 \times 10^6$ and maximum injection time 100 ms) was followed by 10 MS/MS spectra with a resolution of 15,000 of most-abundant peptide ions with a minimum intensity of $4.5 \times 10^4$ that were selected for subsequent collision-induced fragmentation with a stepped collision energy of 27% and an isolation window of 1.3 Da. Fragments were detected by MS/MS acquisition in the orbitrap with a scan range of 200–2000 m/z, automatic gain control target value $5 \times 10^4$, and a maximum injection time 44 ms. Fragmented precursors were dynamically excluded for 15 s.

For antibody-, small molecule-, biomolecule-, and Thanatin-SOG guided LUX-MS samples acquired in DDA mode, peptides were separated by reversed-phase chromatography on an HPLC column (75-μm inner diameter, New Objective) that was packed in-house with a 15-cm stationary phase (ReproSil-Pur C18-AQ, 1.9 μm) and connected to a nano-flow HPLC with an autosampler (EASY-nLC 1000, Thermo Scientific). The HPLC was coupled to a Q-Exactive Plus mass spectrometer (Thermo Scientific) equipped with a nanoelectrospray ion source (Thermo Scientific). Peptides were loaded onto the column with 100% buffer A (99.9% $H_2O$, 0.1% FA) and eluted at a constant flow rate of 300 nl/min with a 70-min linear gradient from 6–28% buffer B (99.9% ACN, 0.1% FA) followed by a 4-min transition from 28 to 50% buffer B. After the gradient, the column was washed for 10 min with 98% buffer B, 4 min with 10% buffer B, and 8 min with 98% buffer B. Electrospray voltage was set to 2.2 kV and capillary temperature to 250 °C. A high-resolution survey mass spectrum (from 300 to 1700 m/z) acquired in the Orbitrap with a resolution of 70,000 at m/z 200 (automatic gain control target value $3 \times 10^6$) was followed by MS/MS spectra in the Orbitrap at a resolution of 35,000 (automatic gain control target value $1 \times 10^6$) of the 12 most-abundant peptide ions with a minimum intensity of $2.5 \times 10^4$ that were selected for fragmentation by higher-energy collision-induced dissociation with a collision energy of 28% and an isolation window of 1.4 Da. Fragmented precursors were dynamically excluded for 30 s.

For the comparison of antibody-SOG LUX-MS (in $H_2O$ and $D_2O$) and antibody-HRP guided proximity labeling, peptides were reconstituted in 3% ACN/0.1% FA/$H_2O$. In all, 1 μg per sample was subjected to liquid chromatography-tandem mass spectrometry using an nLC 1200 (Thermo Scientific) QExactive HFX (Thermo Scientific) setup. Peptides were loaded onto a 30 cm in-house packed column (75 μm ID, New Objective) with mobile phase buffer A containing 0.1% FA/$H_2O$ and separated by reverse-phase chromatography with ReproSil-Pur 120 A C18-AQ 1.9 μm stationary phase (Dr. Maisch GmbH), heated to 50 °C. Peptides were eluted with mobile phase buffer B, consisting of 80% ACN/0.1% FA/$H_2O$, starting from 7% B up to 60% B in 160 min. Peptide intensities were acquired in a data-dependent acquisition mode. MS1 spectra were recorded from 350 to 1650 m/z with a resolution of 60,000 at 200 m/z. MS1 scans were triggered at an AGC target of 3e6 or after 45 ms. Top 12 peptides were iteratively isolated within a 1.3 m/z isolation window and fragmented by high-energy collisional dissociation (HCD) with a normalized collision energy of nce = 28. MS2 scans were triggered upon reaching the AGC target of 1e5 ions or after 22 ms fill time and min AGC of 3e3. Fragment ions were monitored at a resolution of 15,000 at 200 m/z with a first fixed mass of 100 m/z. Fragmented peptides were dynamically excluded for 30 s for further analysis. Ions with charge state unassigned, 1 or >6 were excluded from fragmentation.

For bacteriophage-guided LUX-MS samples acquired in DDA mode, peptides were separated by reversed-phase chromatography on an HPLC column (75-μm inner diameter, New Objective) that was packed in-house with a 15-cm stationary phase (ReproSil-Pur C18-AQ, 1.9 μm) and connected to a nano-flow HPLC with an autosampler (EASY-nLC 1000, Thermo Scientific). The HPLC was coupled to an Orbitrap Fusion Tribrid (Thermo Scientific) equipped with a nanoelectrospray ion source (Thermo Scientific). Peptides were loaded onto the column with 100% buffer A (99.9% $H_2O$, 0.1% FA) and eluted at a constant flow rate of 300 nl/min with a 70-min linear gradient from 6–28% buffer B (99.9% ACN, 0.1% FA) followed by a 4-min transition from 28 to 50% buffer B. After the gradient, the column was washed for 10 min with 98%, 4 min with 10%, and again 8 min with 98% buffer B. Electrospray voltage was set to 1.8 kV and capillary temperature to 275. A high-resolution survey mass spectrum (from 395 to 1500 m/z) acquired in the Orbitrap with a resolution of 120,000 at m/z 200 (automatic gain control target value $2 \times 10^5$ and maximum injection time 50 ms) was followed by MS/MS spectra of most-abundant peptide ions with a minimum intensity of $5 \times 10^3$ that were selected for subsequent

higher-energy collision-induced dissociation fragmentation with a collision energy of 35% and an isolation window of 1.6 Da. Fragments were detected by MS/MS acquisition in the Ion Trap with scan rate set to "Rapid" (automatic gain control target value $1 \times 10^4$ and maximum injection time 250 ms). Fragmented precursors were dynamically excluded for 30 s.

For antibody-guided LUX-MS samples acquired in DIA mode, peptides were separated by reversed-phase chromatography on an HPLC column (75-μm inner diameter, New Objective) that was packed in-house with a 15-cm stationary phase (ReproSil-Pur C18-AQ, 1.9 μm) and connected to a nano-flow HPLC with an autosampler (EASY-nLC 1200, Thermo Scientific). The HPLC was coupled to an Orbitrap Fusion Tribrid mass spectrometer (Thermo Scientific) equipped with a nanoelectrospray ion source (Thermo Scientific). Peptides were loaded onto the column with 100% buffer A (99.9% $H_2O$, 0.1% FA) and eluted at a constant flow rate of 250 nl/min with a 70-min linear gradient from 6 to 28% buffer B (99.9% ACN, 0.1% FA) followed by a 4-min transition from 28 to 50% buffer B. After the gradient, the column was washed for 10 min with 98% buffer B, 4 min with 10% buffer B, and 8 min with 98% buffer B. Electrospray voltage was set to 4.0 kV and capillary temperature to 320 °C. A survey scan acquired in the orbitrap with 120,000 resolution, 50 ms max injection time, and an AGC target of $3 \times 10^6$ was followed by 24 precursor windows with fragmentation at a collision energy of 28% and MS/MS spectra acquisition in the orbitrap with 30,000 resolution. The injection time was set to auto, AGC target to $1 \times 10^6$, and mass range to 300–1700 m/z. For the spectral library, the samples were additionally measured on the Q-Exactive Plus platform in DDA mode as described above

For gp33-guided LUX-MS samples acquired in DIA mode, peptides were separated by reversed-phase chromatography on an HPLC column (75-μm inner diameter, New Objective) that was packed in-house with a 30-cm stationary phase (ReproSil-Pur C18-AQ, 1.9 μm) and connected to a nano-flow HPLC with an autosampler (EASY-nLC 1200, Thermo Scientific). The HPLC was coupled to an Orbitrap Fusion Lumos Tribrid mass spectrometer (Thermo Scientific) equipped with a nanoelectrospray ion source (Thermo Scientific). Peptides were loaded onto the column with 100% buffer A (99.9% $H_2O$, 0.1% FA) and eluted at a constant flow rate of 250 nl/min with a 240-min nonlinear gradient from 4–58% buffer B (80% ACN, 0.1% FA). After the gradient, the column was washed for 10 min with 98% buffer B, 4 min with 1% buffer B. Electrospray voltage was set to 3.6 kV and capillary temperature to 320 °C. A survey scan acquired in the orbitrap with 120,000 resolution, 50 ms max injection time, and an AGC target of $4 \times 10^5$ was followed by 80 precursor windows with fragmentation at a collision energy of 27% and MS/MS spectra acquisition in the orbitrap with 30,000 resolution. The injection time was set to auto, AGC target to $1 \times 10^6$, and mass range to 350–2000 m/z. For the spectral library, replicates of the illuminated and non-illuminated conditions were pooled and analyzed on the same instrumental setup in DDA mode using a universal method with equal gradient and collision energy of 27% and 30%.

**Data analysis.** Raw files acquired in DDA mode were searched against corresponding SwissProt-reviewed protein databases containing common contaminants using Comet (v.2015.01) within the Trans Proteomic Pipeline v.4.7 (SPC/ISB Seattle). Peptides were required to be fully tryptic with a maximum of two missed cleavage sites, carbamidomethylation as fixed modification, and methionine oxidation as a dynamic modification. The precursor and fragment mass tolerance were set to 20 ppm and 1 Da (MS2 ion trap) or 0.02 Da (MS2 orbitrap), respectively. Proteins identified by at least two proteotypic peptides were quantified by integration of chromatographic traces of peptides using Progenesis QI v.4.0 (Nonlinear Dynamics). Contaminant hits were removed, and proteins were filtered to obtain a false discovery rate of <1%. Raw protein abundances were exported based on non-conflicting peptides.

For DIA analysis, raw files for spectral libraries acquired in DDA mode were searched against corresponding SwissProt-reviewed protein databases containing common contaminants using the Sequest HT search engine within Thermo Proteome Discoverer version 2.4 (Thermo Scientific). Peptides were required to be fully tryptic with up to two missed cleavages. Carbamidomethylation was set as a fixed modification for cysteine. Oxidation of methionine and N-terminal acetylation were set as variable modifications. For SILAC data, isotopically heavy labeled arginine (+10 Da) and lysine (+8 Da) were set as variable modifications. Monoisotopic peptide tolerance was set to 10 ppm, and fragment mass tolerance was set to 1 Da (MS2 ion trap) or 0.02 Da (MS2 orbitrap). The identified proteins were assessed using Percolator and filtered using the high peptide confidence setting in Protein Discoverer. Analysis results were then imported to Spectronaut v.13 (Biognosys AG, Schlieren, Switzerland) for the generation of spectral libraries filtering for high confidence identification on peptide and protein level. Raw files acquired in data-independent acquisition mode were analyzed using Spectronaut v.13 with default settings. The proteotypicity filter "only protein group specific" was applied, and extracted feature quantities were exported from Spectronaut using the "Quantification Data Filtering" option. Within the R computing environment (v.3.4.0), protein abundance fold changes (expressed in log2) were calculated using a linear mixed-effect model and tested for statistical significance using a two-sided $t$ test with the appropriate degree of freedom in the R package MSstats (v.3.8.6)[90]. For human proteins, cell surface localization was inferred by matching protein identifications to the in silico human surfaceome[17]. For all other organisms, cell surface localization was inferred based on UniProt GO-term annotations being either "cell surface", "cell membrane", or "secreted". Significantly enriched proteins (abundance fold change >1.5 and $p$ value <0.05) were considered as acute proximity candidates of the ligand- or antibody-SOG construct. The CD20 surfaceome interaction network was generated

using Cytoscape (v.3.7.1)[94]. Protein topology information was retrieved and visualized using the PROTTER interactive webtool[95]. Protein structures were visualized using PyMOL Molecular Graphics System (v.2.4.0, Schrödinger, LLC.). Mass spectrometry data were visualized using ggplot2 (v.3.3.2). Interactive volcano plots were created using an in-house developed R-script using the plotly package (https://plotly-r.com). The 3D-modeled pipette tray of Fig. 1 was created by Thomas Splettstoesser (www.scistyle.com). All other schematic figures were created using the BioRender webtool (https://biorender.com/).

**Reporting summary**. Further information on research design is available in the Nature Research Reporting Summary linked to this article.

## Data availability

The mass spectrometry data generated in this study have been deposited to the ProteomeXchange Consortium (http://proteomecentral.proteomexchange.org) via the PRIDE partner repository[96] under the accession code PXD020481 (LUX-MS data set). SwissProt-reviewed human protein databases were downloaded from https://www.uniprot.org/. Source data are provided with this paper.

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

## Acknowledgements

We acknowledge Julia Boshard, Heidi Funke, and Dr. Patrick Pedrioli for constructive discussions and specific comments on the manuscript. We specifically acknowledge Dr. Damaris Bausch-Fluck and Dr. Andreas Frei for their strategic input and critical discussions in the conceptional stage of the project. We are grateful to the members of the B.W. research group for suggestions and support at all stages of the project. We thank J.R. Wyatt for scientific text editing and T. Splettstoeser for graphical support. We thank Stephan Schneider for the technical assistance on phage enrichment and coupling with SOGs. We further acknowledge Dr. Sebastian Müller, Dr. Patrick Pedrioli, and Dr. Silvana Albert for assistance in analyzing the SILAC-DIA LUX-MS data set and David Vogel for contributing to antibody-guided LUX-MS experiments. The following agencies contributed and are acknowledged for their generous support: ETH (grant ETH-30 17-1 and grant ETH-25 15-2) and Swiss National Science Foundation (grant 31003A_160259) for B.W. and NIH grants R01-GM-094231 and U24-CA210967 for A.I.N.

## Author contributions

M.M. performed all experiments except those noted below. F.G. and N.B. performed isolation and FACS analysis of murine immune cells. Y.S. produced SOG-coupled bacteriophages and performed immunofluorescence experiments. F.W. produced HRP conjugates, performed HRP-based labeling experiments, and performed flow cytometry experiments comparing CD20 abundance. S.N.S. performed transferrin and CD20 labeling experiments. S.U.V. and M.Mo. synthesized SOG-coupled Thanatin. J.R.P. provided CG1 and scientific expertise. Y.Se. performed isolation of PBMC and single-cell chemosensitivity screening. M.v.O. contributed to method optimization. R.H. and R.C.S. synthesized SOG-coupled gp33. A.I.N. contributed analytical tools. M.M., F.W., and B.W. optimized and performed LC-MS/MS acquisition. M.M., F.G., and N.B. analyzed data. M.M., F.G., N.B., Y.S., J.R.P., E.M.C., J.W.B., B.S., J.A.R., M.J.L., A.O., and B.W. designed research. M.M. and B.W. conceived the project and wrote the paper.

## Competing interests

James R. Prudent is an employee/CEO of Centrose LLC, Madison, Wisconsin, USA, a for-profit private company. All other authors declare no competing interests.
