## [Peer Review File · Nature Communications]

REVIEWER COMMENTS

Reviewer #1 (Remarks to the Author):

In this manuscript by Müller et al., the authors describe a light-mediated proximity labeling method for profiling protein-protein interactions on the cell surface. A small-molecule singlet oxygen generator (SOG) is conjugated to antibodies, small molecule drugs and peptide hormones, to achieve specific targeting to the protein of interest on the cell surface. The photosensitizing activity of SOG oxidized nearby proteins, which are subsequently captured by hydrazine-containing probes. The chemical reaction used in this work is distinct from those adopted by other proximity labeling strategies (e.g. APEX, BioID or microMap). Overall, this is a useful technique that could complement existing methods for profiling subcellular proteome with molecular-level spatial precision. The following issues should be addressed before the manuscript is considered for publication:

Major:

1. The authors claimed that LUX-MS offered higher spatial resolution than enzyme-based proteomic profiling techniques (e.g. APEX/HRP) based on an estimation from the life time of reactive intermediates. This should be verified experimentally, for example, by targeting SOG or HRP to the same cell surface receptor protein and comparing the captured protein lists.
2. The authors chose hydrazide as the nucleophilic probe for capturing oxidized histidine without specifying the rationale for this choice. Have they tried probes containing other nucleophilic groups?
3. Related to the above question, the authors claimed observing 2-oxo-histidine as the photooxidation product, but did not provide NMR data or other means of characterization to support the proposed structure. For example, in Supplementary Figure 2, only mass changes were reported. Such information would be important for readers and reviewers to understand the mechanism and to judge the validity of the method.
4. Please specify the light intensities used in this study (W/cm²).
5. Please show the antibody-thiorhodamine conjugation efficiency (number of SOG molecules per antibody).
6. In the case of CG1-SOG, would SOG conjugation cause any changes to drug activity?
7. In Figure 4, a different SOG, methylene blue instead of thiorhodamine was used. The authors should explain the choice of SOG to the readers, and compare their excitation spectrum, protein labeling efficiency, etc..

Minor:

1. In Figure 1 or in Supplementary figures, the chemical transformation from histidine to 2-oxo-histidine upon singlet oxygen generation should be clearly presented.
2. On page 5: "In total, 1674 proteins were quantified by LUX-MS with at least two peptides per protein (Supplementary Table 1)." Please specify whether these are unique peptides?
3. On page 3: "However, in such methods, labeling probes are simultaneously allocated to all surface accessible receptors by antibodies or genetic fusion resulting in a lack of flexibility and specificity to detect ligand-targeted surfaceome signaling domains, leaving the functional hot-spots of the cellular surfaceome landscape unexplored." This statement is somewhat confusing, as the authors also used antibody to direct SOG to specific cell surface protein targets. Please rephrase and clarify.

Reviewer #2 (Remarks to the Author):

This paper from the Wollscheid lab describes LUX-MS - a very useful proximity labeling technique that can be used in a wide range of biological contexts. In several carefully-controlled experiments, the authors show how it can be applied to identify protein "neighbourhoods" on the cell surface, receptors for labeled secreted factors and drugs, bacterial proteins involved in antibiotic and phage binding, and finally T-cell - antigen presenting cell interactions. Overall this is a very impressive study and a useful technique for proximity proteomics. The advantages of this technique appear to be the finely tunable control of the labelling using light. My only substantive criticisms are that the authors could have made more in the examples they selected to emphasize the advantages of LUX-MS over the other available proximity labeling approaches.

Other comments:

Can the authors comment on the specificity of their technique in identifying the targets of the small molecules. The data shown in figure 3c, while identifying known targets (ATP1B3 and BSG) are within a cluster of other targets. Had the targets not been already known, how good would this technique have been at identifying these targets?

Fig. 2c. Why stop at 5 minutes - labeling isn't saturated?

Fig. 5f. Do the authors have better data to reassure the reader that the labelling is indeed cell surface? Flow cytometry of non-permeabilised cells with anti-HA antibody would be more convincing. Also, why are only a fraction of the cells labeled?

Can the authors please double-check the calls to all the figures. I couldn't find the call to Supplementary Fig. 7

Please provide more details on the light source. Readers need to know how they can obtain this if the technique is to be more widely adopted.

Typos:

please check the reference formatting

“in context of”

Fig. 5 legend “pointed by”

Reviewer #3 (Remarks to the Author):

The paper by Muller et al presents LUX-MS, an optoproteomic technology for identifying surface proteins and their interactions. The manuscript is well written and presents an interesting technology for the proximity labeling field. Points to address:

- 1) The authors should characterize sites of protein biotinylation in vivo to determine which amino acids are modified. This can be done by enriching biotinylated peptides using Neutraavidin or an anti-biotin antibody.
- 2) To what extent is the degree of labeling tunable and how does this affect labeling on specific amino acids?
- 3) Why is DIA-MS used to acquire SILAC data presented in Figure 6? Is this necessary to use DIA to implement non-label free quantitation? If not, authors should explain their reasoning.

NCOMMS-20-44287-T | Point-by-point response

Light-mediated discovery of surfaceome nanoscale organization and intercellular receptor interaction networks

REVIEWER COMMENTS

Reviewer #1 (Remarks to the Author):

In this manuscript by Müller et al., the authors describe a light-mediated proximity labeling method for profiling protein-protein interactions on the cell surface. A small-molecule singlet oxygen generator (SOG) is conjugated to antibodies, small molecule drugs and peptide hormones, to achieve specific targeting to the protein of interest on the cell surface. The photosensitizing activity of SOG oxidized nearby proteins, which are subsequently captured by hydrazine-containing probes. The chemical reaction used in this work is distinct from those adopted by other proximity labeling strategies (e.g. APEX, BioID or microMap). Overall, this is a useful technique that could complement existing methods for profiling subcellular proteome with molecular-level spatial precision.

- We would like to thank R1 for the very good questions/feedback which enabled us to clarify statements & claims we made by adding new data and changes to the text/manuscript. The results from some of the experiments we did in response to your requests are super interesting, especially in respect to our recent CD20-related PNAS publication together with my former PhD advisor Michael Reth!
 - K. Kläsener, J. Jellusova, G. Andrieux, U. Salzer, C. Böhrer, S. N. Steiner, J. B. Albinus, M. Cavallari, B. Süß, R. E. Voll, M. Boerries, B. Wollscheid, M. Reth, CD20 as a gatekeeper of the resting state of human B cells, Proc. Natl. Acad. Sci. U. S. A. 118 (2021), doi:10.1073/pnas.2021342118.
 - <https://www.pnas.org/content/118/7/e2021342118>
 - <https://paperpile.com/shared/TtW1qb>

The following issues should be addressed before the manuscript is considered for publication:

Major:

1. The authors claimed that LUX-MS offered higher spatial resolution than enzyme-based proteomic profiling techniques (e.g., APEX/HRP) based on an estimation from the life time of reactive intermediates. This should be verified experimentally, for example, by targeting SOG or HRP to the same cell surface receptor protein and comparing the captured protein lists.

- Given a generic diffusion coefficient, the lifetime of locally generated intermediates correlates with their travelling distance. MicroMap is based on extremely short lived carbene intermediates (nanosecond life time) while APEX/HRP use intermediates with an extended life

time of 0.1 ms to 60 s (Geri et al. 2020). LUX-MS employed singlet oxygen species exert a lifetime of 3.5 μ s in aqueous and cellular systems that can further be extended to 60 μ s in deuterium oxide (heavy water, D₂O) based buffers that are not acutely cytotoxic (Kuimova et al. 2009; Hatz et al. 2007; Esben Skovsen et al. 2005). The labeling range and thus spatial resolution of LUX-MS therefore lies in between MicroMap and APEX/HRP and is furthermore fine-tunable by changing buffer conditions.

- To demonstrate this effect, we added new data to the manuscript and mapped the cell surface microenvironment of CD20 on human resting B-cells (Ramos) using HRP- or SOG-coupled CD20 antibodies and, in addition, performed LUX-labeling in pure H₂O or D₂O. As expected, Ramos cells show a 5-fold lower cell surface expression of CD20 compared to patient-derived human B-lymphoma cells (SUDHL6) (new Supplementary Fig. 4c).

Still, we successfully labeled CD20 and its acute cell surface interaction partners on Ramos cells with all three approaches (new Supplementary Fig. 4d).

- To verify our results, we compared the CD20 surfaceome interaction network of resting Ramos B-cells to constitutively B-cell receptor (BCR) activated SUDHL6 cells. It is known that B-cell receptor (BCR) activation leads to re-localization of CD20 from CD40-associated IgD B-cell receptor clusters to IgM B-cell receptor clusters (Kläsener et al. 2021). Accordingly, we specifically identified Immunoglobulin D (IgD) and found CD40 in the proximity of CD20 on Ramos, but not SUDHL6 cells, even though CD40 is experimentally verified to be located at the cell surface of SUDHL6 cells (Cell Surface Protein Atlas (CSPA) (Bausch-Fluck et al. 2015)). In contrast, Immunoglobulin M (IgM) was largely absent on Ramos (log₂FC 0 - 0.71) but highly

enriched (\log_2FC 4.7) in proximity of CD20 on SUDHL6 cells demonstrating cell-type specific microenvironment mapping of selected cell surface receptors.

- While HRP-based proximity labeling identified >200 CD20 proximity candidates, LUX-MS experiments conducted in the presence of H₂O or D₂O provided both reduced and highly overlapping sets of CD20 proximity candidates - with D₂O yielding additional candidates and improved enrichment of specifically CD20 proximal candidates (**new Fig. 2g** and **new Supplementary Fig. 4e**).

This indicates that, in comparison to HRP-based approaches, cell surface protein proximity mapping by LUX-MS enables cell type specific receptor microenvironment mapping with improved and fine-tunable spatial resolution.

2. The authors chose hydrazide as the nucleophilic probe for capturing oxidized histidine without specifying the rationale for this choice. Have they tried probes containing other nucleophilic groups?

- To design a chemical workflow for the efficient capture of photo-oxidized proteins, we initially explored the protein modification landscape of singlet oxygen (SO) in a discovery-driven fashion using high resolution tandem mass spectrometry and modification-agnostic peptide search using [FragPipe](https://fragpipe.nesvilab.org) (<https://fragpipe.nesvilab.org>). Histidine (His) is known to be the most reactive proteinogenic amino acid towards SO (Kim et al. 2008). Accordingly, we found extensive accumulation of His+14 and His+32 modifications in photo-oxidized transferrin in the initial experiments and in the second set of experiments with varying D₂O buffer conditions performed for this revision (**Supplementary Fig. 1a** and **new Supplementary Fig. 1b**).

- His+14 and His+32 were characterized as 2-oxo-histidine (2-imidazolone) and its hydrated form (Kim et al. 2008; Chang et al. 1997). In addition, oxidized tryptophan and tyrosine were observed. Thus, the photo-oxidation reaction broadly creates carbonyl-containing aldehyde and ketone sites in target proteins. Our lab previously established several technologies to profile the surfaceome and extracellular interactions (autoCSC, TRICEPS-LRC, HATRIC-LRC) that rely on oxidation of sialic acids of cell surface glycoproteins into aldehyde derivatives for subsequent capture using hydrazide-containing probes (van Oostrum et al. 2019; Sobotzki et al. 2018; Frei et al. 2012; Wollscheid et al. 2009). Hydrazide probes are widely established for stable and selective labeling of carbonyl-functionalized proteins under physiological conditions (pH 5-7). We therefore tested and confirmed the ability of these probes to biotinylate photo-oxidized proteins (Supplementary Fig. 1c) and implemented the use of a novel catalyst (2-(aminomethyl)imidazole dihydrochloride) to boost labeling efficiency (Supplementary Fig. 4b). This enabled us to leverage on previous experience and rapidly built an optimized chemical workflow for efficient profiling of cell surface receptor neighborhoods leaving the necessity to screen for additional probes with distinct nucleophilic groups behind.

3. Related to the above question, the authors claimed observing 2-oxo-histidine as the photooxidation product, but did not provide NMR data or other means of characterization to support the proposed structure. For example, in Supplementary Figure 2, only mass changes were reported. Such information would be important for readers and reviewers to understand the mechanism and to judge the validity of the method.

- Histidine is described as the most reactive proteinogenic amino acid towards singlet oxygen-mediated oxidation (Kim et al. 2008; Dahl et al. 1988; Matheson and Lee 1979). The photo-oxidation products of free histidine ((Agon et al. 2006; Uchida and Kawakishi 1993)) or histidine in proteins such as cytochrome c (Kim et al. 2008), lysozyme (Marques et al. 2017), glucose-6-phos-phate dehydrogenase (G6PDH) (Leinisch et al. 2017) and antibodies (igG) (Amano et al. 2014) were extensively characterized using electrochemical detection (HPLC-ECD), amino acid analysis, ¹⁶O- and ¹⁸O-labeling, Matrix-assisted laser desorption/ionization time-of-flight mass spectrometry (MALDI-TOF MS), intact as well as bottom-up electrospray ionization tandem mass spectrometry (ESI-LC-MS/MS), electron paramagnetic resonance (EPR) and nuclear magnetic resonance (NMR) spectroscopy. The extensive accumulation of

data illustrates 2-imidazolone (2-oxo-histidine) and its hydrated form as major products emerging from a dynamic and highly heterogeneous background of intermediates and crosslinked species (reviewed here (Di Mascio et al. 2019)). Using ultra high-resolution tandem mass spectrometry (mass resolution of 120'000 both at precursor ion level) and high confidence peptide identification criteria (false discovery rate < 1%) we obtain over 5600 spectra with fragments specifically matching to histidine modified peptides (data uploaded to ProteomeXchange Consortium). We further show their dependency on photo-oxidation and reactivity towards a hydrazide-containing biotin linker underlining the light-induced generation and utilization of 2-oxo-histidine (2-imidazolone) and its hydrated form for the capture of cell surface labeled protein neighborhoods within the LUX-MS framework.

- To increase clarity for the reader, we revised the corresponding section “*Development of the LUX-MS technology*” in the main text accordingly.

4. Please specify the light intensities used in this study (W/cm²).

- For the light-activated receptor microenvironment mapping on the surface of living cells, we used high power Precision LED spotlights controlled by a BioLED Light Control Module in continuous wave mode (Mightex Systems, Pleasanton, USA). For LUX-MS applications with thiorhodamine as singlet oxygen generator (SOG), we used spotlight BLS-PLS-0590-030-05-S for illumination at 590 nm with a light intensity of 4.6 mW/cm². For LUX-MS applications with methylene blue as singlet oxygen generator (SOG), we used spotlight BLS-PLS-0656-030-07-S for illumination at 656 nm with a light intensity of 14.9 mW/cm².
- For clarification, we added a separate subsection “*Light sources*” to the methods description in the main text.

5. Please show the antibody-thiorhodamine conjugation efficiency (number of SOG molecules per antibody).

- To couple antibodies with thiorhodamine SOGs for subsequent LUX-MS experiments, we leveraged on a study that optimized the functionalization of a broad spectrum of (commercially) available antibodies using NHS chemistry (van Buggenum et al. 2016). Using a 5-fold molar excess of NHS-functionalized probes thereby lead to predominant labeling of surface accessible primary amines of the heavy chain and efficient production of functional antibody conjugates.
- To investigate the average degree of labeling (DOL) of antibody-SOG constructs we established a protocol for absorbance-based DOL determination. Specifically, absorbance of conjugates, antibodies (Ab) and free thiorhodamine (SOG) at both 280 and 582 nm was measured using a Nanodrop 2000 spectrophotometer (Thermo Scientific) and DOL calculated as described previously (Marshall et al. 2016). Incubation of IgG isotype control antibodies with equimolar, 5-fold or 10-fold excess of thiorhodamine for 1 h at room temperature resulted in a DOL of 0.5, 1.5 or 2, respectively (**new Supplementary Fig. 2a**).

Given the overall high structural similarity between the IgG subclasses IgG1 - IgG4 (over 90% homology in amino acid sequence with most variability stemming from the non-accessible hinge region (Vidarsson et al. 2014)), we expect an average of 1.5 thiorhodamine molecules per antibody in all our antibody-guided LUX-MS applications.

- We added this information to section “LUX-MS enables proteome-wide mapping of antibody binding targets and surfaceome nanoscale organisation” in the main text.

6. In the case of CG1-SOG, would SOG conjugation cause any changes to drug activity?

- We thank the reviewer for highlighting this point. Unravelling the cell surface interactions of small molecule drugs such as CG1 is important for drug discovery and development. By utilizing a small molecular SOG probes instead of macromolecular enzymes, such as APEX/BioID etc, LUX-MS now enables small molecule directed microenvironment mapping on the surface of living cells. To highlight the compatibility of small molecules to LUX-employed SOGs over enzymes, we coupled the cytotoxic drug CG1 (0.5 kDa) to thiorhodamine SOG (0.7 kDa) or HRP (44 kDa) and performed a single cell chemosensitivity screen using human acute promyelocytic leukemia (HL60) cells. Using our absorbance-based DOL protocol we confirmed the coupling of CG1 to aldehyde-activated HRP and determined an average of 2.5 CG1 molecules per HRP (**new Supplementary Fig. 6a**). Treatment of HL60 cells with >0.1 μ M CG1 and CG1-SOG lead to significant drop in cell count after 24 hours (50% decrease, **new Supplementary Fig. 6b and c**) and 48 hours (75% decrease, **new Fig. 3b**). In contrast, no effect was observed for CG1-HRP even at 1 μ M and after 48 hours. In summary, while coupling of HRP essentially abolished CG1 drug activity in the concentration range tested, CG1-SOG retained drug activity with reduced potency ($IC_{50}(\text{CG1-SOG}) = 93 \text{ nM}$ compared to $IC_{50}(\text{CG1}) = 3.5 \text{ nM}$) and enabled small molecule directed receptor identification and microenvironment mapping using LUX-MS.

- We added this information to section “LUX-MS decodes surfaceome signaling domains of small molecule drugs and biomolecules” in the main text and moved Figure 1b to the supplement as **Supplementary Fig. 6d**.

7. In Figure 4, a different SOG, methylene blue instead of thiorhodamine was used. The authors should explain the choice of SOG to the readers, and compare their excitation spectrum, protein labeling efficiency, etc..

- While an incredible diversity of singlet oxygen generating reagents exist that could potentially be implemented into the LUX-MS framework, small molecule SOGs are most promising to unlock new application spaces due to their photochemical properties and ligand compatibility (as demonstrated above). Methylene blue (MB) is a widely applied photosensitizer in photobiology and photodynamic therapy and was chosen for initial LUX-MS experiments due to its favorable singlet oxygen quantum yield (Φ^1O_2) of 0.52 and absorbance maximum (λ_{max}) at 668 nm with molar extinction coefficient (ϵ_{max}) of $1.1 \times 10^5 \text{ M}^{-1}\text{cm}^{-1}$ (Tardivo et al. 2005). However, MB has a low fluorescence quantum yield (Φ_F) of 0.04, is easily reduced in biological environments and is prone to formation of dimers with λ_{max} shifted to 590 nm and diminished generation of singlet oxygen in favor of superoxide and other reactive oxygen species (Tardivo et al. 2005). In the course of this work, we thus switched to thiorhodamine, a small molecule SOG with a lower Φ^1O_2 of 0.21, an λ_{max} at 582 nm and ϵ_{max} of $1.1 \times 10^5 \text{ M}^{-1}\text{cm}^{-1}$ but a 10-fold higher Φ_F of 0.44 (Holt et al. 2006) eventually allowing for fluorescence microscopy-based imaging of subcellular SOG distribution. Furthermore, to the best of our knowledge, the aggregation of thiorhodamines to multimers with distinct photochemical

properties has not been described indicating enhanced stability and specificity for the generation of singlet oxygen species. The expected two-fold decrease in protein labeling efficiency when switching from MB to thiorhodamine (two-fold lower Φ^1O_2) could further be abrogated by the use of D_2O , hydrazone-formation catalysts and prolonged illumination times. The extensive library of diverse singlet oxygen generators and the vast tuning capabilities of the photo-labeling reaction therefore provide LUX-MS the flexibility to be tailored to a wide range of specific research applications.

Minor:

1. In Figure 1 or in Supplementary figures, the chemical transformation from histidine to 2-oxo-histidine upon singlet oxygen generation should be clearly presented.

- We modified Figure 1 to also include the chemical transformation from unmodified histidine to 2-oxo-histidine and its suggested biotinylated form in SOG-proximal proteins after illumination (**new Fig. 1**).

2. On page 5: "In total, 1674 proteins were quantified by LUX-MS with at least two peptides per protein (Supplementary Table 1)." Please specify whether these are unique peptides?

- If not otherwise stated, only proteotypic peptides (i.e. peptides matching to one specific protein) were considered of which at least two unique peptides (i.e. unique combination of peptide sequence and charge state) were required for a protein to be relatively quantified.
- We added this information to section "LUX-MS enables proteome-wide mapping of antibody binding targets and surfaceome nanoscale organisation" and methods section in the main text.

3. On page 3: “However, in such methods, labeling probes are simultaneously allocated to all surface accessible receptors by antibodies or genetic fusion resulting in a lack of flexibility and specificity to detect ligand-targeted surfaceome signaling domains, leaving the functional hot-spots of the cellular surfaceome landscape unexplored.” This statement is somewhat confusing, as the authors also used antibody to direct SOG to specific cell surface protein targets. Please rephrase and clarify.

- With this statement, we intended to highlight the current inability to specifically resolve ligand-targeted surfaceome signaling domains with available antibody- or genetics-based methodologies. We performed antibody-guided experiments as proof-of-principle and for global surfaceome nanoscale mapping approaches but extended the use the LUX-MS technology to unravel surfaceome signaling domains of small molecules, biomolecules, intact viruses and interacting immune cells clearly breaking said analytical boundaries. Still, for clarity we rephrased the sentence of the introduction in the main text accordingly:
- *“While such antibody-based methods are useful in deciphering surface microenvironments of targeted receptors, novel and highly versatile approaches are required to also uncover surfaceome domains that underlie the signaling function of small molecules, biomolecules, viral particles and complex intercellular receptor interaction networks as formed in the immunosynapse of interacting immune cells during T-cell activation.”*

Reviewer #2 (Remarks to the Author):

This paper from the Wollscheid lab describes LUX-MS - a very useful proximity labeling technique that can be used in a wide range of biological contexts. In several carefully-controlled experiments, the authors show how it can be applied to identify protein “neighborhoods” on the cell surface, receptors for labeled secreted factors and drugs, bacterial proteins involved in antibiotic and phage binding, and finally T-cell - antigen presenting cell interactions. Overall, this is a very impressive study and a useful technique for proximity proteomics. The advantages of this technique appear to be the finely tunable control of the labelling using light. My only substantive criticisms are that the authors could have made more in the examples they selected to emphasize the advantages of LUX-MS over the other available proximity labeling approaches.

- We thank the reviewer for the overall positive feedback and for pointing out the broad applicability of the LUX-MS technology. Indeed, in the experiments shown, we primarily focused on understanding the photochemical basis of the LUX labeling reaction, it’s fine-tunability using light, D₂O, pH and catalysts and how it can be leveraged to provide valuable insights in distinct biological scenarios of the vast life science disciplines. In order to emphasize the advantages and complementary benefits of LUX-MS, we performed additional experiments directly comparing the labeling range and ligand-compatibility of small molecule SOG-based LUX-MS to HRP enzyme-based proximity labeling approaches. We performed receptor proximity mapping on living human B-cells using both SOG- and HRP-coupled antibodies against CD20. We thereby found LUX-MS to provide a set of reasonable CD20 proximal candidates that could be extended by the use of D₂O but that was generally narrower

compared to HRP (**new Fig. 2g**) indicating light-activated receptor microenvironment mapping with improved and fine-tunable spatial resolution.

- Furthermore, we attempted to perform small molecule guided proximity labeling on living cells for target receptor identification and surfaceome neighborhood mapping using both small molecule SOG and HRP enzymes. However, coupling of HRP to the cytotoxic small molecule drug CG1 abolished CG1 drug activity even at concentrations of 100-fold its IC₅₀(CG1) and after 48 hours (**new Fig. 3b** and **new Supplementary Fig. 6c**).

- In contrast, coupling of the small molecule SOG thiorhodamine to CG1 retained its drug activity and enabled small molecule-directed identification of the target receptor and its surfaceome microenvironment using LUX-MS. We hope with these additional sets of data contrasting spatial resolution and ligand compatibility we could emphasize the advantages of LUX-MS over other available proximity labeling approaches.

Other comments:

Can the authors comment on the specificity of their technique in identifying the targets of the small molecules. The data shown in figure 3c, while identifying known targets (ATP1B3 and BSG) are within a cluster of other targets. Had the targets not been already known, how good would this technique have been at identifying these targets?

- In contrast to direct ligand-receptor interaction screening technologies (such as TRICEPS-based LRC or HATRIC-based LRC)(Frei et al. 2012; Sobotzki et al. 2018), LUX-MS not only reveals small molecule targeted receptors but also the immediate microenvironment/nanoscale organization in which they reside in on the surface of living cells. The LUX-MS technology thereby provides an additional layer of spatial information/spatial proteotyping that aids in the formulation of potential molecular mechanisms of action (MMoA) and in the rational design of proximity-enhanced drugs with improved specificity (i.e. drug-conjugated DutaFabs(Beckmann et al. 2021), bispecific antibodies(Kontermann 2012) or extracellular-drug conjugates (EDC)(Marshall et al. 2016)).
- For applications primarily focused on target identification, the specificity of LUX-MS can eventually be tuned towards target receptors by modulating the labeling range (adapting light and buffer conditions) or by using additional controls such as a ligand competition control to eliminate ligand-independent proximity candidates (please see for reference the Thanatin experiment, Fig. 4). Furthermore, normalizing observed light-dependent enrichments to protein-specific potentials of getting labeled (e.g., based on the number and accessibility of extracellular histidine) would refine the enrichment-to-proximity translation and enable to identify the most SOG-proximal and therefore ligand-interacting target receptors.

Fig. 2c. Why stop at 5 minutes - labeling isn't saturated?

- It is well known that lipids of the cellular membrane can undergo oxidation by singlet oxygen making them susceptible to biotinylation by biocytin-hydrazide. The biotinylation observed by fluorescence-activated cell sorting (FACS) is independent of the nature of labeled biomolecules and therefore a poor readout to estimate cell surface protein labeling saturation. We therefore relied on light-dependent enrichment of proteins quantified by high-resolution tandem mass spectrometry and found a well-defined cluster of CD20 and known interacting proteins that was significantly enriched after 5 min of illumination (**Supplementary Fig. 2c** and **Fig. 2f**). Longer labeling times may enhance sensitivity on the cost of spatial and temporal resolution, but likely result in the chemical degradation of proteins for extended periods.

Fig. 5f. Do the authors have better data to reassure the reader that the labelling is indeed cell surface? Flow cytometry of non-permeabilised cells with anti-HA antibody would be more convincing. Also, why are only a fraction of the cells labeled?

- To demonstrate that LUX-MS technology can be used to profile/decode cell surface interactions of intact viruses with spatial specificity, we set out to verify the cell surface location of two selected proximity candidates (BN389_06470 and BN389_05780) that were identified by phage-guided LUX-MS on the gram-positive bacteria *Listeria monocytogenes* (LM). There is until today no experimental evidence confirming the surface localization of either protein. The gene of the latter one encodes a putative ABC transporter which is membrane associated and likely buried in the cell wall(Rismondo and Schulz 2021). In the initial experiment shown in the manuscript, we therefore relied on an immunostaining protocol that is well-established to detect LM surface proteins through the use of a cell wall

degrading lysozyme(Sumrall et al. 2019). To reassure the cell surface specificity of the imaging data we performed a second experiment with an adapted immunostaining protocol that stains live and entirely untreated LM bacteria to exclude any potential effect of lysozyme treatment on bacterial cell wall integrity. The current availability of instruments and local safety policies would require harsh antibiotic treatment of stained bacteria for flow cytometric analysis, eventually leading to unreliable results in terms of subcellular localization. We therefore again used the in-house available confocal microscope for single-cell fluorescence detection. The obtained results reconfirm the surface-accessibility of selected candidates on intact LM bacteria and therefore validate the spatial specificity of LUX-MS to profile cell surface interactions of intact viruses (new Fig. 5f).

- Concerning the labeling pattern, selected candidate proteins were fused with the HA-tag while retaining the native promoter, meaning the expression of tagged proteins is dependent on endogenous regulation and growth phase. This translates into cell state-specific labeling patterns and lower signal-to-noise ratios compared to the artificial and constitutive overexpression using engineered promoters. In fact, the endogenous expression system and subcellular resolution of microscopy allow for identification of labeling hotspots at the cell poles i.e. of BN389_05780-HA cells indicating polarized protein distribution that is known to be of functional importance for other LM proteins such as ActA to direct actin-based cell motility(Smith et al. 1995).

Can the authors please double-check the calls to all the figures. I couldn't find the call to Supplementary Fig. 7

- Figure numbering was fixed and supplementary figures were condensed to increase clarity and conciseness of the manuscript.

Please provide more details on the light source. Readers need to know how they can obtain this if the technique is to be more widely adopted.

- We agree and added a separate subsection "Light sources" to the methods description in the main text with the following description of the model numbers, specifications and providers of all light sources used in this work:

- “Photo-oxidation reactions were controlled using Precision LED spotlights operated via a BioLED Light Control Module (BLS-PL04-US) in continuous wave mode (Mightex Systems, Pleasanton, USA). For LUX-MS applications with thiorhodamine as singlet oxygen generator (SOG), spotlights BLS-PLS-0590-030-05-S were used for illumination at 590 nm with a light intensity of 4.6 mW/cm². For LUX-MS applications with methylene blue as singlet oxygen generator (SOG), spotlights BLS-PLS-0656-030-07-S were used for illumination at 656 nm with a light intensity of 14.9 mW/cm².”

Typos:

please check the reference formatting

“in context of”

Fig. 5 legend “pointed by”

- Corrected.

Reviewer #3 (Remarks to the Author):

The paper by Muller et al presents LUX-MS, an optoproteomic technology for identifying surface proteins and their interactions. The manuscript is well written and presents an interesting technology for the proximity labeling field.

- We thank the reviewer for the feedback/comments and we are confident that the LUX-MS technology will have a significant impact in decoding ligand receptor interactions, for surfaceome discovery and mapping surfaceome nanoscale organization across organisms.

Points to address:

1) The authors should characterize sites of protein biotinylation in vivo to determine which amino acids are modified. This can be done by enriching biotinylated peptides using Neutravidin or an anti-biotin antibody.

- Given the success of methods such as DiDBiT(Schiapparelli et al. 2014) or BioSiTe(Kim et al. 2017) in identifying for example APEX-biotinylated peptides(Udeshi et al. 2017) using mass spectrometry, we initially intended to quantitatively trace the light-dependent formation of biotin-labeled peptides of LUX-MS using high resolution tandem mass spectrometry (LC-MS/MS). Based on the chemical structure and reaction mechanism of our biotin linker (biocytin-hydrazide, BH), we defined and screened for peptides with BH modifications in a closed search LC-MS analysis of photo-labeled transferrin peptides. However, while oxidative amino acid modifications could readily be detected, no biotinylated peptides were identified initially. We then employed MSfragger, an ultra-fast search engine enabling open modification search modes (i.e. the identification of peptides with unspecified modifications)(Kong et al. 2017). The software was extremely successful in *de novo* mapping the amino acid modification landscape introduced by photosensitized singlet oxygen, but did not reveal peptides with additional masses of around 380 Da (e.g. the mass of a single biocytin-hydrazide linker) (**new**

Supplementary Fig. 1). We hypothesized that in contrast to modifications by NHS-biotin, biotinoyl-5'-AMP (BioID) or biotin-phenol (APEX), functionalization of a peptide with the long, membrane-impermeable biotin-hydrazide linker renders them non-detectable by mass spectrometry for example by lowering their ionization efficiency. We therefore took a different approach, by measuring the depletion of photo-oxidized transferrin peptides upon addition of BH allowing us to identify amino acid modifications that serve as light-dependent protein biotinylation sites. Future developments of LUX-MS may implement the use of a cleavable biotin linker for the release of streptavidin captured peptides enabling precise identification of protein sites biotinylated during the light-triggered proximity labeling event.

2) To what extent is the degree of labeling tunable and how does this affect labeling on specific amino acids?

- As described above, we initially uncovered the amino acid modification landscape of photosensitized singlet oxygen in transferrin proteins with and without BH using high resolution LC-MS/MS combined with open modification searching using MSfragger. We thereby identified previously characterized oxidation products of amino acids methionine, cysteine, tyrosine, tryptophan and in particular histidine of which His+14 Da, His+32 Da and Cys+48 Da showed most significant reactivity towards BH (**Supplementary Fig. 1c**). The photosensitization and lifetime of singlet oxygen is well known to be dependent on environmental conditions such as temperature, pressure, solvent viscosity, D₂O content and pH (Fresnadillo and Lacombe 2016). For example, replacing H₂O with D₂O in the buffer system extended the lifetime of singlet oxygen by 20-fold (Kuimova et al. 2009) providing an attractive means to tune the spatial resolution and specificity of the LUX-MS approach. To investigate the effect of D₂O on the light-controlled generation of oxidative modifications and potential protein labeling sites, we performed the photo-oxidation of transferrin for 15 min at increasing D₂O/H₂O ratios and relatively quantified peptides across conditions using high resolution LC-MS/MS. Results showed overall similar modification profiles with substantial production of BH-reactive His+14 and His+32, increasing production of Cys+48, Tyr+16 and consumption of unmodified peptides indicating D₂O-enhanced production of biotin labeling sites (**new Supplementary Fig. 1b**).

- Accordingly, performing antibody-guided LUX-labeling on living cells in D₂O increased the degree of light-induced cell surface biotinylation 4-fold compared to H₂O (**new Supplementary Fig. 4b**) resulting in the identification of additional and potentially more SOG-distance receptor (new Fig. 2e). Furthermore, increasing the pH is known to boost photo-oxidation of amino acids especially histidine (Matheson and Lee 1979) and directly translated into a 1.8-fold increase in cell surface biotinylation when moving from pH 6 to 7.4 (**new Supplementary Fig. 4b**).

Thus, multiple handles exist to fine-tune the degree of light-controlled labeling giving LUX-MS the flexibility to be tailored to answer research questions in a broad range of scientific disciplines.

3) Why is DIA-MS used to acquire SILAC data presented in Figure 6? Is this necessary to use DIA to implement non-label free quantitation? If not, authors should explain their reasoning.

- In the Stable Isotope Labeling by Amino acids in Cell culture (SILAC) approach, proteins are metabolically labeled in vivo with heavy-isotope containing amino acids and pooled with an unlabeled (light) sample for combined sample preparation and mass spectrometric analysis (Ong et al. 2003). While the method is typically employed for the accurate relative quantification of proteins between two conditions, we used SILAC to isotopically barcode distinct cell types (i.e. T-cells: Light, dendritic cells: Heavy) prior antigen-guided LUX-MS allowing us to achieve proteotype mapping within functional immunosynapses with unrivaled cell-type specificity (**Fig. 6**). SILAC in combination with data-dependent acquisition mass spectrometry (SILAC-DDA) is well-established in the field. However, a recent study demonstrated that combining SILAC with data-independent acquisition - a recently developed acquisition mode that systematically measures peptides with enhanced comprehensiveness, reproducibility and accuracy of quantification - results in dramatic improvement in quantitative accuracy and precision by an order of magnitude (Pino et al. 2021). SILAC-DIA is thereby able to capture minimal fold changes of low abundant proteins due to fewer missing values in the quantification matrix as compared to SILAC-DDA. Considering spatially restricted protein labeling within a limited number of immunosynaptic connections by LUX-MS, SILAC-

DIA is for us the method of choice for discovery-driven elucidation of intercellular receptor interaction networks within functional immunosynapses on a proteome-wide scale.

REVIEWERS' COMMENTS

Reviewer #1 (Remarks to the Author):

The authors have exhaustively engaged in responding to comments raised by all referees. The newly added data comparing HRP and SOG are particularly interesting. I recommend publication in Nature Communications.